# Extensive rewiring of the EGFR network in colorectal cancer cells expressing transforming levels of KRAS$^{G13D}$

Susan A. Kennedy et al.[#]

Protein-protein-interaction networks (PPINs) organize fundamental biological processes, but how oncogenic mutations impact these interactions and their functions at a network-level scale is poorly understood. Here, we analyze how a common oncogenic KRAS mutation (KRAS$^{G13D}$) affects PPIN structure and function of the Epidermal Growth Factor Receptor (EGFR) network in colorectal cancer (CRC) cells. Mapping >6000 PPIs shows that this network is extensively rewired in cells expressing transforming levels of KRAS$^{G13D}$ (mtKRAS). The factors driving PPIN rewiring are multifactorial including changes in protein expression and phosphorylation. Mathematical modelling also suggests that the binding dynamics of low and high affinity KRAS interactors contribute to rewiring. PPIN rewiring substantially alters the composition of protein complexes, signal flow, transcriptional regulation, and cellular phenotype. These changes are validated by targeted and global experimental analysis. Importantly, genetic alterations in the most extensively rewired PPIN nodes occur frequently in CRC and are prognostic of poor patient outcomes.

[#]A full list of authors and their affiliations appears at the end of the paper.

PPINs are a major principle of biological organization. Several large-scale studies have demonstrated their importance in organizing fundamental cellular processes[1–4]. Increasing evidence suggests that PPINs are altered in human disease and that such changes contribute to pathogenesis[3,5,6]. However, we lack a clear comprehension of how such changes occur and how they affect PPIN function. This particularly applies to oncogenic mutations, where we rarely understand how system wide effects are generated. For instance, oncogenic RAS mutations, which activate the ability of RAS proteins to engage downstream effectors, occur in ~30% of all human cancers[7]. RAS mutated cancers are resistant to most targeted therapies, and inhibiting downstream effectors has proven ineffective, likely because of complex feedback structures in the downstream pathways and the large number of effector pathways[8]. These impediments highlight the need for a systems level understanding of the RAS signaling network and the changes associated with RAS transformation[8]. The $KRAS^{G13D}$ mutation ($mtKRAS^{G13D}$) investigated here is the second most frequent RAS mutation in CRC[7], and is associated with aggressive behavior and poor clinical outcomes[9].

We use quantitative mass spectrometry (qMS) to map KRAS regulated PPINs in two closely related CRC cell lines that express either transforming or non-transforming levels of $mtKRAS^{G13D}$. Focusing on the epidermal growth factor receptor (EGFR) signaling network, where KRAS plays a key role in CRC[10], we analyze 1710 immunoprecipitates and map >6000 PPIs involved in EGFR signaling (Fig. 1). To analyze this dataset we develop an analysis pipeline for the quantitative comparison of PPIN data between different cell lines. Finding that the expression of transforming levels of mtKRAS correlates with substantial rewiring of the EGFR PPIN, we analyze the functional consequences of this rewiring on protein complex assemblies, information flow, and biological responses including the prognosis of CRC patients. To facilitate the utilization of this extensive data for further research we develop PRIMESDB.eu (https://primesdb.eu/), an integrated database and analysis platform for exploring the PPINs described here.

## Results
**Mapping the EGFR PPIN in mtKRAS Cells**. We mapped the effects of mtKRAS on PPINs downstream of the EGFR in HCT116 cells, which have been widely used to study mtKRAS functions in CRC. HCT116 harbor an oncogenic $KRAS^{G13D}$ allele, which was targeted for disruption by homologous recombination to generate the non-tumorigenic HKE3 cells[11]. A thorough genetic, biochemical and biological characterization described in a previous publication[12] and Supplementary Fig. 1 confirmed that HCT116 and HKE3 are closely related cell lines. Compared to HCT116 cells, HKE3 had a non-transformed phenotype, as reflected by EGFR inhibitor sensitivity and reduced migration, proliferation, colony forming ability, and anchorage independent growth (Supplementary Fig. 1). Interestingly, despite this non-transformed phenotype HKE3 cells retain a genetically stable low-level expression of $mtKRAS^{G13D}$, likely due to a duplication of the mutant $mtKRAS^{G13D}$ allele in HCT116 and knockout of only one copy in the HKE3 cells (Supplementary Fig. 1A). Recent findings indicate that oncogenic KRAS mutations occur in normal tissues and that KRAS activity needs to exceed a threshold to drive cancer progression and metastasis[13–16]. Thus, this cell line pair offers the opportunity to compare the EGFR PPIN in cells expressing a transforming vs. a non-transforming dosage of mtKRAS. Furthermore, HCT116 are not addicted to mtKRAS for survival, minimizing selection pressure to acquire compensatory mutations when mtKRAS dosage is reduced[17].

To attain a representative coverage of the EGFR PPIN we selected 95 bait proteins (Supplementary Data 1) based on a highly curated EGFR signaling network map[18] and a literature survey of the EGFR pathways involved in CRC pathogenesis and progression. The baits cover the main functions of EGFR signaling including key kinases, phosphatases, scaffold and adapter proteins in the network. The baits were expressed as FLAG-tagged proteins carefully titrating transfection to achieve a similar expression level in both cell lines. Baits were immunoprecipitated and associated prey proteins were identified by high-resolution Orbitrap qMS. To ensure high data quality, we analyzed 95 bait and empty vector control immunoprecipitates (IPs) from both forward and reverse SILAC labeled cells using three biological and two technical replicates per bait resulting in 1710 samples and 1140 qMS analyses (Fig. 1). As common analysis methods for AP-MS data are ill suited for quantitatively comparing PPI data from different cell lines and from a biased bait selection, we used HiQuant[19], which we specifically developed analyse MS data from complex experiments such as ours. The pipeline includes rigorous steps for data quality control, normalization, statistical analysis, and network construction. This workflow includes a stringent two-step procedure to exclude false positive interactors (Supplementary Methods, section 12). The EGFR PPINs in the HCT116 and HKE3 cells, termed EGFR-Net$^{mtKRAS-Hi}$ and EGFRNet$^{mtKRAS-Lo}$, respectively, were reconstructed from high-confidence bait–prey interactions.

**EGFR PPIN network architecture**. EGFRNet$^{mtKRAS-Hi}$ and EGFRNet$^{mtKRAS-Lo}$ consist of 3162 and 2788 bait-prey interactions, respectively (Supplementary Fig. 2A–E, Supplementary Data 2). This network size is within the expected distribution of known PPINs (Supplementary Fig. 3A). 93 of the 95 baits had a least one prey detected in both cell lines. More than 70% of preys in EGFRNet$^{mtKRAS-Lo}$ were also nodes in EGFRNet$^{mtKRAS-Hi}$, indicating that most nodes are true components of the EGFR PPIN, since they were independently detected in both cell lines. Both EGFRNets are small-world, single-component networks (i.e., all nodes are reachable from one another) with similar scale-free topologies and comparable other network properties such as average path length, node degree, betweenness centrality (bc), and clustering coefficients (Supplementary Fig. 2F–I). For example, many of the major hubs (highly connected nodes) in EGFR-Net$^{mtKRAS-Hi}$ are also hubs in EGFRNet$^{mtKRAS-Lo}$ including GRB2, RAB5A, RAF1, and SH2D3C. While GRB2 is a well-known hub coordinating different aspects of EGFR signaling[20], some of the other high bc nodes are thought to have specialized functions. Our data suggest that these proteins are involved in a far greater degree of crosstalk with other cellular processes than previously assumed.

A comparison with currently known PPIs shows that >80% of interactions discovered in our study are new, attesting to the value of focused PPIN mapping studies complementing genome-wide efforts and suggesting that many PPIs may be highly dependent on the cellular context. Such a high proportion of novel interactions is consistent with other large-scale, AP-MS based, interactome mapping efforts[1,2,4,21,22]. Testing 17 arbitrarily chosen interactions using conventional co-immunoprecipitation/Western blot experiments showed that the PPI data were highly reproducible by a different method (Supplementary Fig. 3B, C).

As AP-MS may underrepresent interactors of integral membrane proteins[23], we used MYTH, a membrane yeast two-hybrid assay[24], to identify binary protein interactors of the human EGFR family, ERBB1–4 (Supplementary Methods, section 30). This interactome map comprised 405 interactions (Supplementary Fig. 4 and Supplementary Data 3) including 181 new interactions. All bait–prey interactions detected in theEGFRNets can be explored in the Supplementary Data and at http://primesdb.eu/.

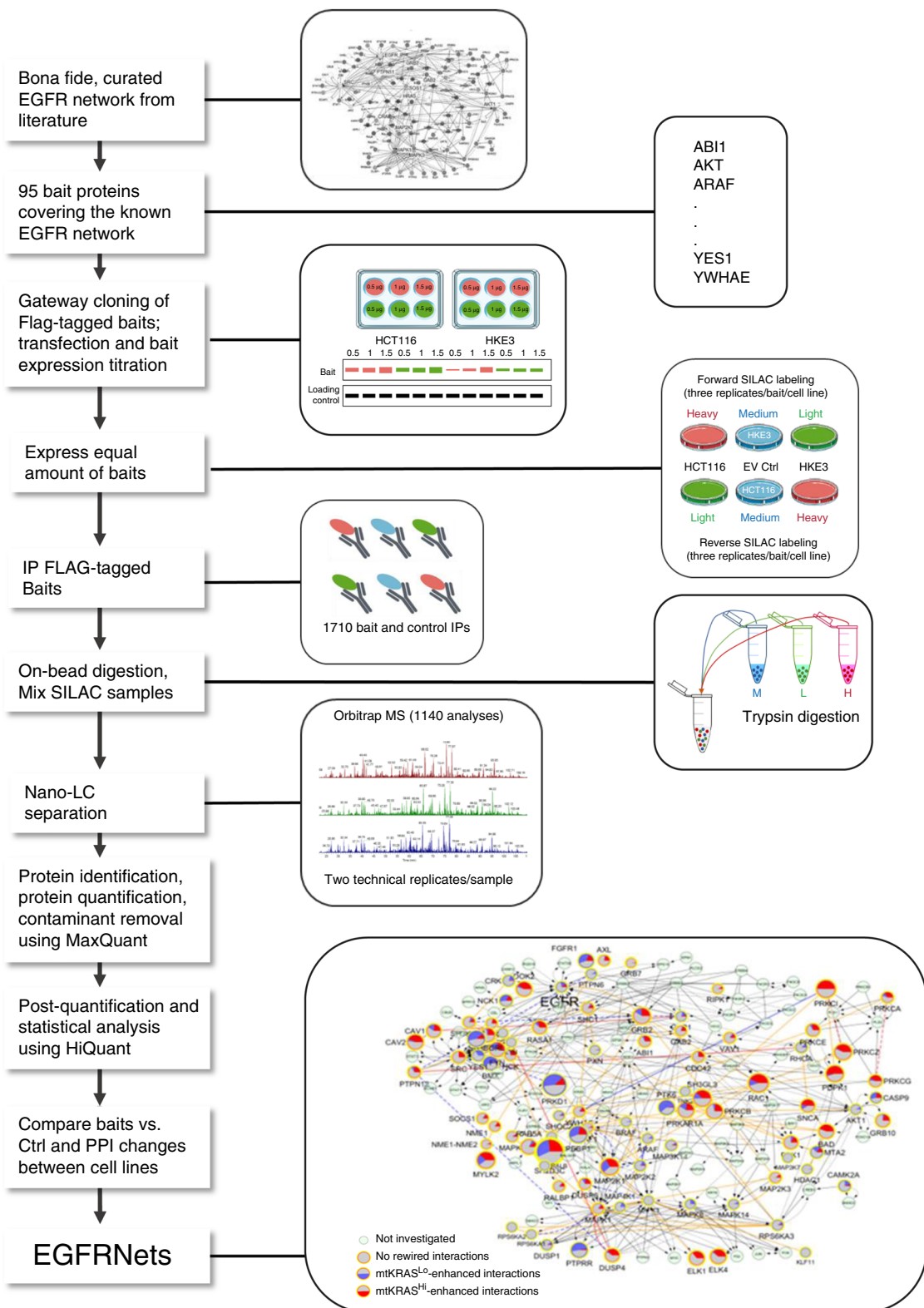

**Fig. 1 Experimental and data analysis workflow for the comparative mapping of PPIs in the EGFR network.** Baits were chosen based on the core EGFR network described by Kiel et al.[18] and additional manually curated literature information. Flag-tagged expression vectors were constructed using the Gateway cloning system and transfected into HCT116 (mtKRAS$^{Hi}$) and HKE3 (mtKRAS$^{Lo}$) cells. Careful titration of the transfected plasmids ensured similar protein expression in both cell lines grown in SILAC media as monitored by Western blotting. For MS experiments similar amounts of baits were expressed in SILAC labeled HCT116 and HKE3 cells and immunoprecipitated (IP) with anti-Flag antibodies. To assure robust quantitation the SILAC label was swapped, i.e. each bait was isolated from HCT116 and HKE3 cells grown in heavy or light medium, respectively. After trypsin digestion peptides were identified and quantified by orbitrap mass spectrometry. Raw data were analysed using MaxQuant[59] and further processed using HiQuant[19] implementing a stringent pipeline to retain only true interactors. Based on these data two quantitative protein–protein interaction networks, termed EGFRNet$^{mtKRAS-Hi}$ and EGFRNet$^{mtKRAS-Lo}$, were reconstructed and are shown in a combined differential network representation. EV Ctrl, empty vector control transfection.

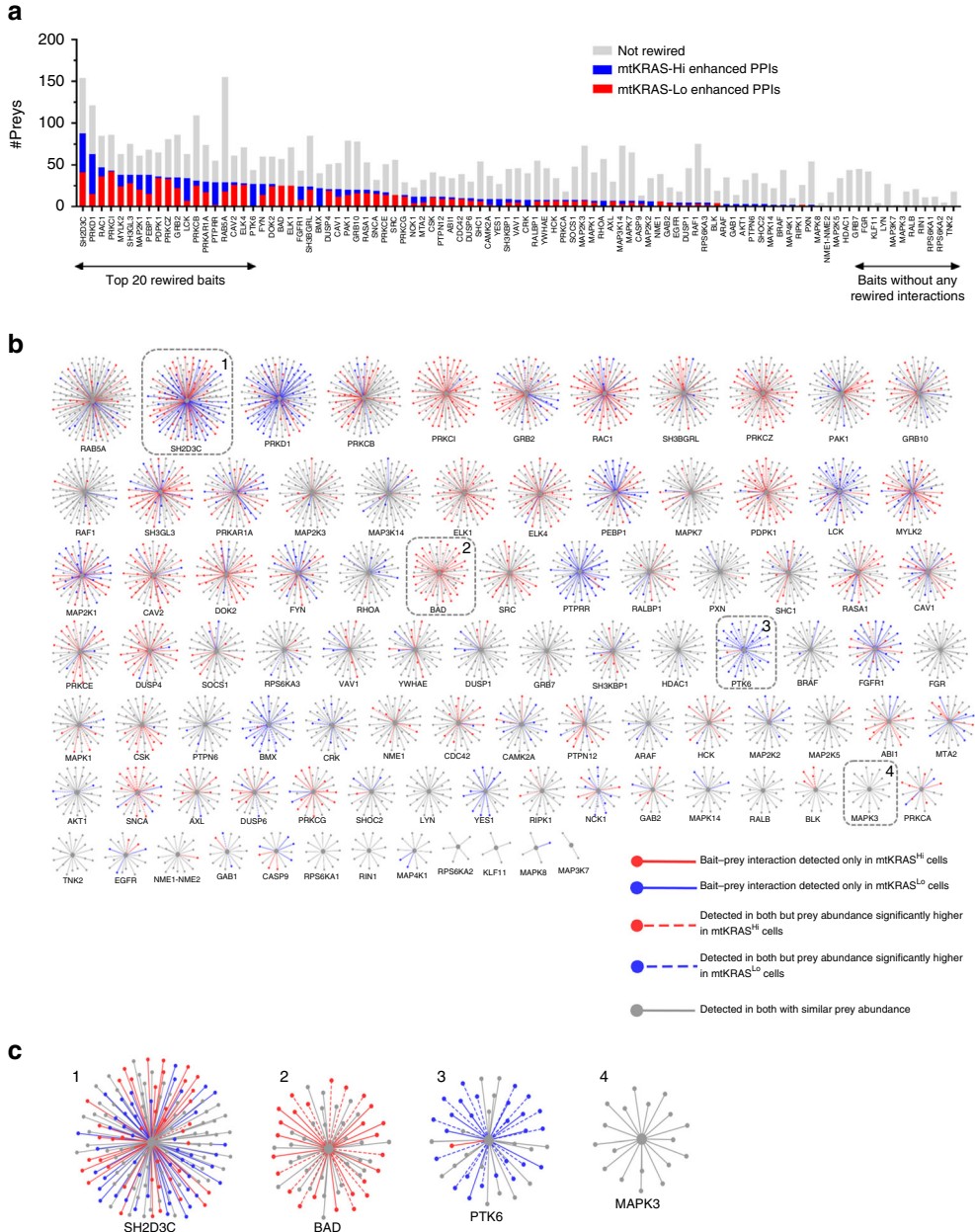

**Fig. 2 The EGFRNet^mtKRAS-Hi and EGFRNet^mtKRAS-Lo PPINs are rewired. a** The number of preys identified for each bait-prey AP-MS complex. Red, rewired preys enhanced in mtKRAS^Hi cells; blue, rewired prey proteins enhanced in mtKRAS^Lo cells. AP-MS complexes are named based on the bait protein and are shown on the *x*-axis. **b** Network spoke model view of rewired interactions. Bait–prey interactions identified in both networks with similar prey abundance are shown in gray. Bait–prey interactions that were identified only in EGFRNet^mtKRAS-Hi or EGFRNet^mtKRAS-Lo are shown as solid red or blue edges, respectively. Bait–prey interactions that were detected in both networks but where prey abundance was significantly higher in EGFRNet^mtKRAS-Hi or EGFRNet^mtKRAS-Lo are shown as dotted red or blue edges, respectively. Visit primesdb.eu to explore the complexes in more detail/with greater resolution. **c** Zoom-in on four nodes that represent mixed, preferential or no rewiring. Source data are provided as a Source Data file.

**mtKRAS^Hi induces extensive PPIN rewiring**. To identify interactions that were significantly rewired in mtKRAS^Hi cells, i.e., interactions that were gained/lost in one EGFRNet or present in both but with significantly altered prey abundance, we statistically compared prey abundance between each bait-prey complex in the EGFRNets. Of the 4420 bait–prey interactions detected in at least one EGFRnet (Supplementary Data 4), 1368 were significantly rewired i.e., prey abundance was significantly different between the two PPINs at $P \leq 0.05$, significance $A \leq 0.05$ (Supplementary Data 5). Six hundred and thirty four of the rewired interactions were edges only in EGFRNet^mtKRAS-Hi, and 406 were edges only in EGFRNet^mtKRAS-Lo indicating that most rewiring is due to interaction gains or losses (Fig. 2). The 328 remaining rewired interactions were present in both networks but with significantly different prey abundances ($P \leq 0.05$, significance $A \leq 0.05$). These data suggest that the oncogenic mtKRAS activity in mtKRAS^Hi cells initiates a ripple effect throughout the network substantially altering network topology far beyond direct KRAS interactors.

**Potential drivers of PPIN rewiring**. To investigate which molecular mechanisms could drive PPIN rewiring, we first analyzed whether genetic mutations other than *KRAS^G13D* played a role, since genetic variation has previously been associated with

PPIN rewiring[25]. Using whole genome sequencing we identified genetic alterations, including copy number variations (CNVs), insertions/deletions (InDels), synonymous and nonsynonymous single-nucleotide-variants (SNVs) between the two cell lines (Supplementary Data 6–8; Supplementary Fig. 5A). Using the Genome Analysis Toolkit[26] 27 genes were predicted to be impacted by structural variants, but no gene was a node in the EGFRNets. Considering CNVs, five genes were EGFRNet nodes, but only one gene product, PPP3CA, was rewired. Of the 170,135 SNVs and small InDels found different between mtKRAS[Hi] and mtKRAS[Lo] cells 1091 were variants of predicted high/medium impact[27] (Supplementary Data 6). Of these, 70 were nodes in the EGFR PPI network and 36 were rewired. Considering that EGFRnets contain 4420 nodes, of which 1360 have rewired interactions, SNVs affect 1.6% of nodes and 2.6% of rewired interactions. These data suggest that structural variants, SNVs and CNV-driven changes in gene/protein expression have limited impact on EGFRNet rewiring. Nonetheless, we cannot rule out that these or other genetic differences influence some PPIs by affecting gene promoter usage, mRNA editing, or codon usage. We also considered that rewired prey could simply represent lowly or highly expressed nodes. However, we found no bias in the gene expression distribution of rewired nodes compared to unchanged nodes (Supplementary Fig. 5B) suggesting that genetic changes that alter gene/protein expression, e.g., CNVs, do not make major contributions to PPIN rewiring.

To further explore this, we directly tested whether changes in protein expression between the two cell lines are linked to the observed EGFRNet rewiring. We profiled protein abundances in the mtKRAS[Hi] and mtKRAS[Lo] cell lines using qMS (Supplementary Data 9). 404 of the 4685 proteins quantified showed a significant difference in abundance ($P \leq 0.05$). Pathway analysis revealed that proteins more abundant in mtKRAS[Hi] were enriched for roles in the cell cycle, consistent with the increased proliferation rate of these cells (Supplementary Fig. 1E). By contrast, proteins more abundant in mtKRAS[Lo] cells were enriched for roles in oxidative phosphorylation, lipid metabolism, and the lysosome. The decreased expression of proteins involved in oxidative phosphorylation in mtKRAS[Hi] cells is consistent with a metabolic switch from oxidative phosphorylation to glycolysis, a hallmark of cancer cells known as the Warburg effect[28]. A strong relationship between *KRAS[G12D]* dosage and increased glycolysis was recently reported[29]. Similarly, lipid metabolism reprogramming is also a hallmark of cancer cells, including CRC cells[30].

We found a weak ($r^2 = 0.18$) but significant correlation ($P < 0.001$) between fold-change in abundance in the AP-MS protein complexes and fold-change in protein expression between the cell lines (Supplementary Fig. 5C). One hundred and fourteen differentially expressed (DE) proteins were nodes in the EGFRNets (Fig. 3a, Supplementary Fig 5D, E), two of them corresponding to baits (RPS6KA1 and SH3KBP1, $\Delta = 1.7$-fold). This was not more than statistically expected ($P = 0.054$) indicating that the EGFR network was not especially enriched for DE proteins. However, 74 of the 114 DE proteins represented rewired nodes, which was statistically significant ($P = 4.21E-5$), confirming an association between differential node abundance and network rewiring. Interestingly, some bait-prey complexes were particularly enriched for DE proteins (Supplementary Fig. 5F, G). For example, of 71 rewired preys in the SH2D3C complex, 16 (22%) were also DE. Overall, these data suggest that differences in protein expression between mtKRAS[Hi] and mtKRAS[Lo] cells may underlie some of the rewired interactions. However, this association was lost when considering proteins at DE >2-fold. Furthermore, as ~90% of rewired nodes were not identified as DE proteins, differences in protein expression

alone cannot explain the wide-spread network rewiring. We have not investigated the reverse possibility that PPIs may affect protein stability[31,32].

To assess other potential drivers of PPIN rewiring, we examined protein phosphorylation, which can generate docking sites and affect binding affinities between proteins, thereby influencing protein complex formation. qMS-based phosphoproteome analysis of mtKRAS[Hi] and mtKRAS[Lo] cells identified 384 differentially phosphorylated proteins (Supplementary Data 10). Two hundred and seventy one proteins were preferentially phosphorylated in mtKRAS[Hi] cells and were enriched for roles in cell cycle and apoptosis related pathways, while the 121 proteins preferentially phosphorylated in mtKRAS[Lo] cells were weakly enriched for cytokine signaling and related processes (Supplementary Data 10). Eighty nine differentially phosphorylated (DP) proteins were nodes in the EGFRNets (Fig. 3b). Compared to the number of network nodes that were also represented in the phosphoproteomics screen, this was not more than statistically expected ($P = 0.06$) indicating that the EGFRNets were not enriched for DP proteins. However, 56 of the 89 DP proteins (63%) mapping to EGFRNets were also significantly rewired nodes ($P < 0.01$). Interestingly, this association was even stronger when considering interactions that were enhanced (or only found) in EGFRNet[mtKRAS-Hi]. Rewiring of these interactions was significantly correlated with higher phosphorylation in the mtKRAS[Hi] cells ($P < 0.001$). These results suggest that differential phosphorylation could contribute to PPIN rewiring, especially when mtKRAS signaling is high (Supplementary Fig. 5F, G). However, as with differentially expressed proteins, the majority of rewired nodes were not associated with differential phosphorylation. This suggests that the observed network rewiring is an emergent property of the changed cellular state in cells expressing transforming levels of mtKRAS and is not readily predicted by changes in any single factor such as protein expression or phosphorylation.

**PPIN rewiring modifies protein complexes and their functions**. Pathway analysis of all 735 prey proteins involved in rewired interactions revealed a statistically significant enrichment for roles in processes including RNA splicing, mitochondrial translational, protein folding by the chaperonin-containing-TCP1 (CCT) complex and cell migration (Fig. 3c, Supplementary Data 11). Interestingly, CRISPR and shRNA-based screens of HCT116 cells and isogenic wild-type *KRAS* derivatives found that synthetic lethal *KRAS[G13D]* genes had roles in mRNA splicing and mitochondrial translation, and that these processes were required for *KRAS[G13D]* oncogenicity[33]. Proteins encoded by synthetic lethal genes identified in this CRISPR screen were significantly enriched (9 of 55; $P < 0.01$) for rewired PPIs in our network. The shRNA screen in this study identified several CTT components as synthetic lethal genes. In our study, several baits co-precipitated the entire CCT complex in both mtKRAS[Hi] and mtKRAS[Lo] cells (Supplementary Data 4), with interactions between CCT and the baits BMX, LCK, and PTK6, being significantly rewired (Supplementary Data 5).

Similar correlations were observed when assessing how PPIN rewiring affected the composition of known protein complexes described in CORUM, a curated database of experimentally determined mammalian protein complexes[34]. We detected 42 and 40 CORUM complexes where at least 70% of their component proteins were nodes in EGFRNet[mtKRAS-Hi] and EGFRNet[mtKRAS-Lo], respectively (Supplementary Fig. 6, Supplementary Data 12). Several of these CORUM complexes are involved in processes mediated by EGFR signaling including actin cytoskeleton organization, RAF, and NFκB signaling[35,36].

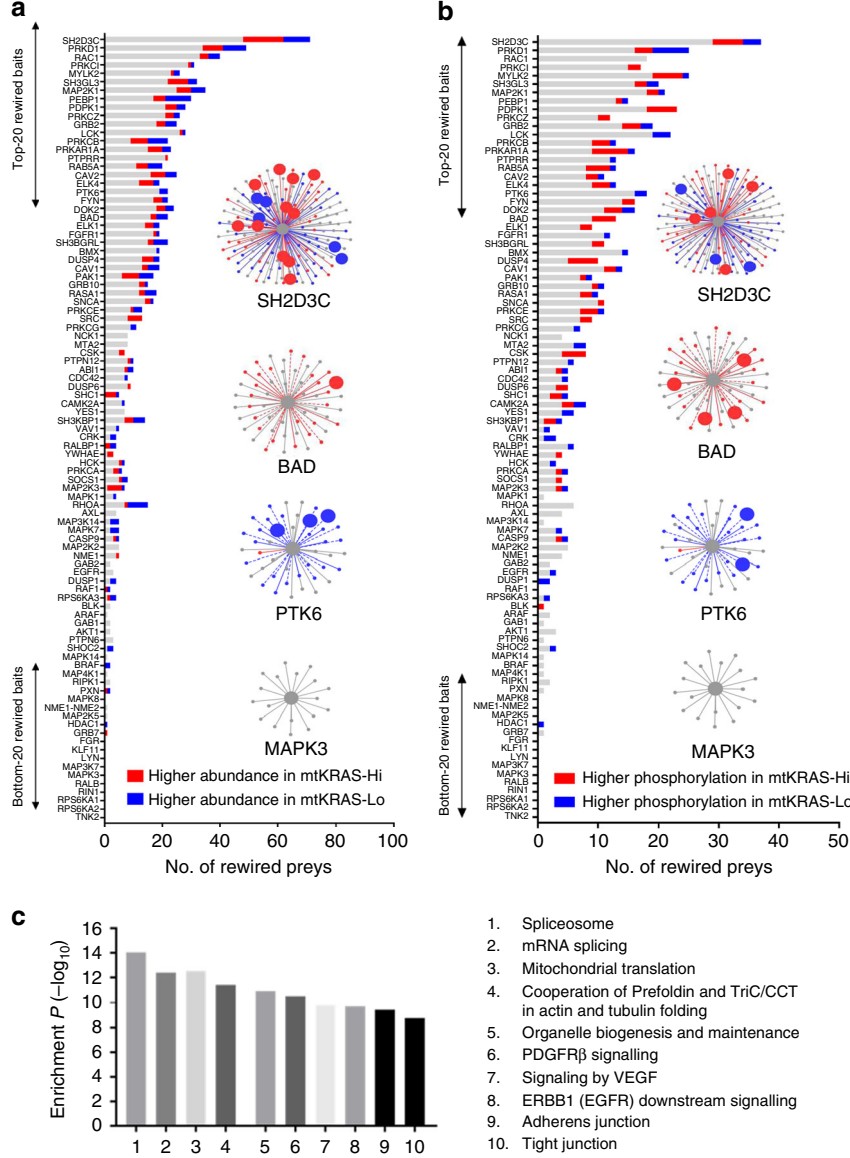

**Fig. 3 Potential drivers of the EGFR PPI network rewiring. a** The number of rewired prey proteins for each bait-prey AP-MS complex that were assessed for differential protein expression between the two cell lines. Rewired prey proteins that were significantly more abundant in the mtKRAS[Hi] cells are shown in red. Rewired prey proteins that were significantly more abundant in the mtKRAS[Lo] cells are shown in blue. Four selected AP-MS complexes highlighting differentially abundant prey proteins (larger nodes) are also shown. **b** The number of rewired prey proteins for each bait-prey AP-MS complex that were assessed for differential phosphorylation between the two cell lines. Rewired prey proteins that were significantly more phosphorylated in the mtKRAS[Hi] cells are shown in red. Rewired prey proteins that were significantly more phosphorylated in the mtKRAS[Lo] cells are shown in blue. Four selected AP-MS complexes highlighting differentially phosphorylated prey proteins (larger nodes) are also shown. **c** Statistically enriched pathways among the 735 prey proteins involved in rewired interactions. Source data are provided as a Source Data file.

However, other complexes participate in functions not usually associated with EGFR signaling, e.g. chromatin modification, regulation of protein folding, mRNA splicing and protein translation. This analysis suggests that PPIs organize different aspects of EGFR signaling including roles that have not been characterized yet. Most CORUM complexes were present in both networks, although some were extensively rewired. CORUM complexes, where >60% of constituent proteins were rewired preys, included complexes involved in mRNA transcription, splicing and protein folding (Supplementary Fig. 6) suggesting that such house-keeping functions support mtKRAS transformation. Furthermore, complex formation can stabilize proteins[31] and may contribute to the differential protein abundance between mtKRAS[Hi] and mtKRAS[Lo] cells.

Interestingly, rewired interactions were non-randomly distributed across the bait-prey complexes (Supplementary Fig. 7). Many bait-prey complexes, including AKT1, FGR, HDAC1, LYN, MAPK3 (ERK1), RIPK1, and TNK1 complexes, had no significantly rewired interactions, while others, such as MAP2K1 (MEK1), RAC1, and SH2D3C, were substantially rewired (Supplementary Data 5). In some cases, rewiring predominantly involved the gain of new bait–prey interactions in mtKRAS[Hi] cells. For example, all 25 rewired bait–prey interactions in the BAD complex were only detected or significantly more abundant in the EGFRNet[mtKRAS-Hi] (Fig. 4a). BAD is a proapoptotic protein[37], which contributes to the higher apoptosis rate of mtKRAS[Hi] vs. mtKRAS[Lo] cells, as shown by siRNA knockdown experiments (Fig. 4b). Interestingly, the BAD interactions

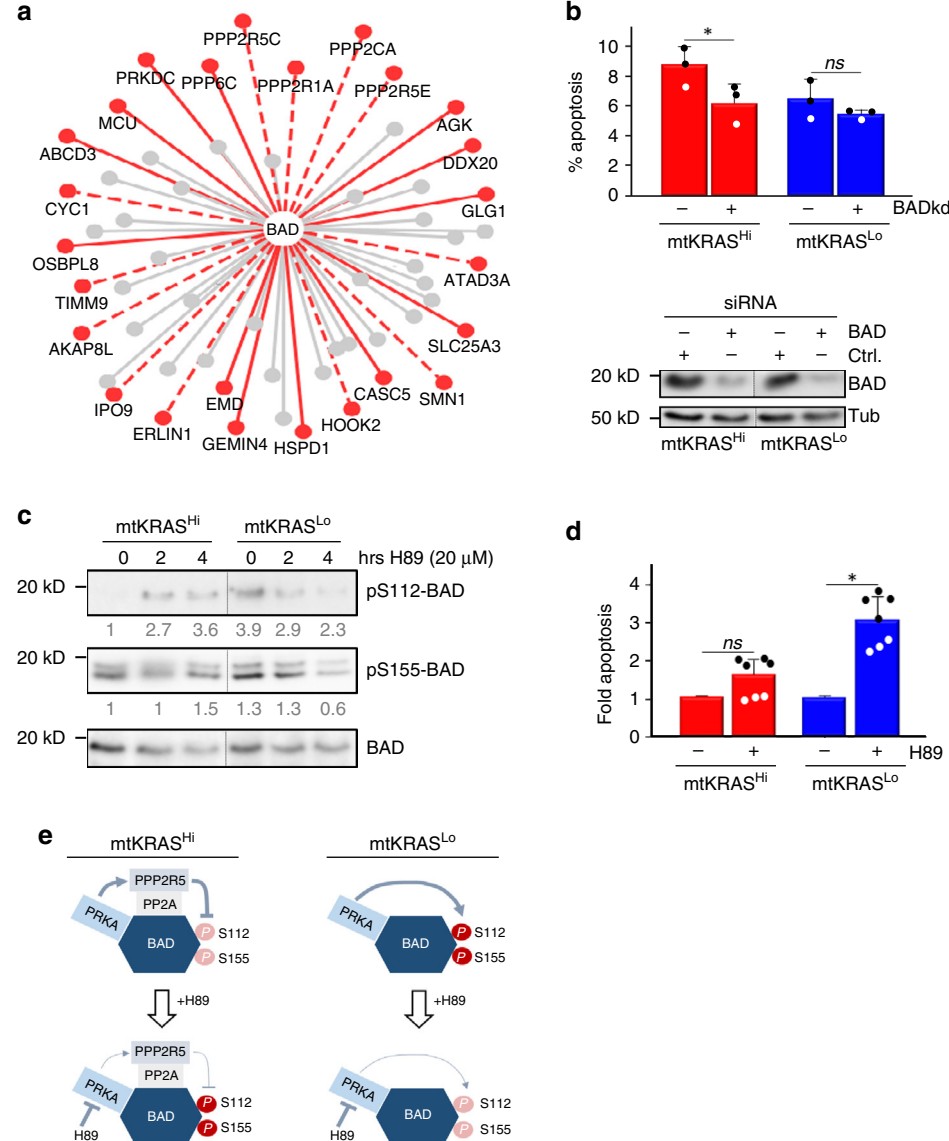

**Fig. 4 Differential regulation of BAD protein phosphorylation and its biological effect. a** PPI interactions in the BAD complex. Red broken lines, PPIs enhanced in EGFRNet$^{mtKRAS-Hi}$ vs. EGFRNet$^{mtKRAS-Lo}$; red solid lines, EGFRNet$^{mtKRAS-Hi}$ exclusive interactions; gray, unchanged interactions. **b** Knocking down BAD expression significantly reduces apoptosis in mtKRAS$^{Hi}$ but not mtKRAS$^{Lo}$ cells. Apoptosis was measured 24 h post treatment. Ctrl., untargeted siRNA; BADkd, BAD specific siRNA. The reduction of BAD protein expression was assayed by Western blotting using tubulin (Tub) as loading control. Apoptosis assays represent three independent experiments; error bars are SD, and * means significant ($P < 0.05$) according to Student's $t$-test; ns, not significant. **c** mtKRAS$^{Hi}$ and mtKRAS$^{Lo}$ cells were treated with the PRKA inhibitor H89 as indicated before BAD phosphorylation at S112 and S155 were assessed by Western blotting using phospho-specific antibodies. Numbers below lanes represent BAD phosphorylation normalized to total BAD protein expression. Samples shown in **b** and **c** are from the same Western blots, where irrelevant lanes were removed as indicated by vertical lines. **d** Apoptosis in response to 20 μM H89 treatment was assessed as in **a**. The data represent two independent experiments with 3 and 4 biological replicates, respectively. Error bars are SD, and * means significant ($P < 0.05$) according to Student's $t$-test; ns, not significant. **e** A proposed model of the differential biological effect of PRKA due to differential PPIs. Source data are provided as a Source Data file.

enhanced in EGFRNet$^{mtKRAS-Hi}$ included a higher abundance of protein phosphatase 2A (PP2A) family members (Supplementary Data 5), which correlated with a lower phosphorylation of BAD on S112 and S155 in HCT116 (Fig. 4c). These sites inactivate BAD's pro-apoptotic function and can be phosphorylated by cAMP dependent protein kinase (PRKA)[37], which interacted with BAD in both cell lines. Inhibition of PRKA reduced S112 and S155 phosphorylation in mtKRAS$^{Lo}$ cells, while PRKA inhibition increased BAD phosphorylation in mtKRAS$^{Hi}$, especially on S112 (Fig. 4c). This differential action of PRKA is likely due to its ability to activate PP2A by phosphorylating the B56δ subunit (PPP2R5)[38], which is mainly bound to BAD in mtKRAS$^{Hi}$ cells.

Consequently, PRKA inhibition preferentially enhanced apoptosis in mtKRAS$^{Lo}$ cells (Fig. 4d). These results suggest that PPI rewiring can profoundly subvert the biological effects of PRKA signaling, in this case converting a survival signal into a pro-apoptotic signal (Fig. 4e).

Another substantially rewired node was PTK6 (Supplementary Fig. 8). PTK6 is a poorly characterized tyrosine kinase, which is amplified or overexpressed in 16% of CRC patients. PTK6 can stimulate CRC cell survival and oncogenic signaling in a kinase dependent manner, but suppresses epithelial-to-mesenchymal transition in a kinase independent fashion[39]. PTK6 rewiring mostly decreased interactions with the CCT chaperonin complex

in mtKRAS$^{Hi}$ cells, whereas the interaction with metastasis associated 1 family member 2 (MTA2) was increased (Supplementary Fig. 8A). Given the key role that MTA2 plays in cell motility and metastasis[40], we examined whether PTK6 contributed to the differential cell migration observed between mtKRAS$^{Hi}$ and mtKRAS$^{Lo}$ cells (Supplementary Fig. 1F). mtKRAS$^{Hi}$ and mtKRAS$^{Lo}$ expressed similar amounts of endogenous PTK6 (Supplementary Fig. 8B). Overexpressing PTK6 accelerated migration in HCT116 cells but inhibited it in HKE3 cells (Supplementary Fig. 8C). Increased migration was dependent on PTK6 kinase activity, as a kinase dead PTK6 mutant slowed migration (Supplementary Fig. 8D). Knocking down endogenous PTK6 decreased migration specifically in mtKRAS$^{Hi}$ but not in mtKRAS$^{Lo}$ cells (Supplementary Fig. 8E). Taken together, these results suggest that PTK6 preferentially enhances migration in cells with high KRAS activity.

**Network rewiring alters information flow through EGFRNets.** The extensive changes in network wiring and protein complex composition suggested that PPIN rewiring may alter signal processing in the EGFR network. First, we explored how different concentrations of active KRAS affects the formation of KRAS complexes with known effector proteins. Activated RAS proteins signal by binding a range of effectors through a single, shared binding domain[7] leading to competition between effectors. To analyze the formation of specific KRAS-effector complexes we constructed an equilibrium binding model of proteins competing for a single target (see Methods section). This model classifies KRAS effectors into low and high affinity binders, whose binding dissociation constants ($K_d$'s) are greater or smaller, respectively, than the abundance of active KRAS. It shows that for low-affinity effectors the corresponding KRAS complex concentrations are proportional to the effector concentration divided by the $K_d$, whereas for high-affinity interactors the resulting KRAS complexes concentrations are determined by the abundance of active KRAS and effectors alone. Thus, changes in mtKRAS concentration can profoundly rearrange the composition of KRAS-effector complexes, which rather than changing the strengths of downstream pathway activation shifts signaling from high to low affinity effectors as mtKRAS dosage increases (Fig. 5a). As measured by quantitative Western blotting, the mtKRAS concentrations in mtKRAS$^{Lo}$ and mtKRAS$^{Hi}$ cells are ~150 nM and ~400 nM, respectively, indicating that high affinity RAS effector complexes prevail in mtKRAS$^{Lo}$ cells, while low affinity effectors dominate signaling in mtKRAS$^{Hi}$ cells. Specifically, the model predicted that fold-changes in KRAS-bound fractions are higher for low-affinity than for high-affinity effectors. Ranking effectors by the fold-changes in KRAS-bound fractions allowed us to estimate their relative contribution to downstream signaling. Applying this analysis to baits that participate in bona-fide KRAS effector pathways (RAF/MAPK, RAL, PI3K, TIAM, AFDN, PLCε, and RIN1), we calculated the sensitivity of a node responding to different mtKRAS doses by summing the log fold-changes of interactions (normalized by the number of pathway nodes measured) in each bait-prey AP-MS complex. These experimentally deduced sensitivity ranks of KRAS-effector complexes correlated with the model-predicted ranks (Supplementary Data 13). These results suggest that the threefold difference in mtKRAS activity induces the formation of very different KRAS-effector complexes that initiate network rewiring by engaging different signaling pathways (rather than stronger activate the same set) that propagate changes further downstream.

Given these extensive changes in network wiring and protein complex composition we hypothesized that PPIN rewiring would also alter signal processing, leading to differential activation of downstream transcriptional programs. In order to investigate this hypothesis in an unbiased way not limited to known KRAS effector pathways, we employed a computational modeling based approach called information flow (IF) analysis[41,42]. This method simulates IF in a network through discrete-time random walks from a source node, i.e., EGFR, to downstream sinks, i.e., transcription factors (TFs). To model the impact of PPIN rewiring, we simulated IF independently in the EGFRNet$^{mtKRAS-Hi}$ and EGFRNet$^{mtKRAS-Lo}$ networks and calculated an IF score (IFS) for each node in the two networks that reflects the volume of signals flowing through a node. Nodes with high IFS in both networks included known key transducers of EGFR signaling, e.g., GRB2[43], indicating that major hubs are used regardless of mtKRAS dosage. However, 119 nodes had a >2-fold difference in IFS in EGFRNet$^{mtKRAS-Hi}$ vs. EGFRNet$^{mtKRAS-Lo}$ (Fig. 5b and Supplementary Data 14), indicating potentially critical differences in signal processing. Interestingly, many of the highest scoring nodes that received more information flow in the EGFRNet$^{mtKRAS-Hi}$ network were proteins involved in protein folding including heat shock protein (HSP) 70 family members (HSPA1A and HSPA8), and HSP90 family members (HSP90AA1, HSP90AB1, and HSP90AB3P). HSP70 and HSP90 expression is upregulated in many cancers including CRC[44], and high HSP70 expression is associated with poor clinical outcomes in CRC[45]. Furthermore, HSP90 inhibitors are in clinical trials for several cancers including CRC[46]. Another node with higher IFS in EGFRNet$^{mtKRAS-Hi}$ was SRC, which is a major promoter of CRC proliferation, metastasis, drug resistance, and is over-expressed in ~80% of CRCs[47]. These data suggest that network nodes with increased IF in mtKRAS$^{Hi}$ cells contribute to the molecular pathogenesis of CRC and may represent potential drug targets.

Next, we assessed whether PPIN rewiring alters IF to transcription factors (TFs) in the EGFRNets (Fig. 5c). FOXO1 and MYC were predicted to receive higher IF in EGFRNet$^{mtKRAS-Hi}$. Assessing gene expression in both cell lines by RNAseq prior to and following EGFR activation by TGFα, revealed that *FOXO1* and *MYC* were more highly expressed in HCT116 cells (Fig. 5d, e). On the other hand, TFs including STAT1 and FOS received higher IF in EGFRNet$^{mtKRAS-Lo}$, and their gene expression was significantly elevated in HKE3 cells (Fig. 5f, g). These results were consistent with the IF model predictions. Next, we analyzed TF binding sites in the promoters of genes that were differentially regulated between mtKRAS$^{Hi}$ and mtKRAS$^{Lo}$ cells. Genes upregulated in mtKRAS$^{Hi}$ cells were enriched for MYC binding sites (Fig. 5h), consistent with the prediction that MYC receives more IF through EGFRNet$^{mtKRAS-Hi}$. Conversely, genes upregulated in mtKRAS$^{Lo}$ were enriched for the interferon-stimulated response element (ISRE) motif (Fig. 5i), a key motif in the promoters of STAT1/2-regulated genes[48]. The difference in *FOS* gene expression between the cell lines was particularly evident 60 min post-TGFα stimulation. Consistent with the prediction of higher FOS regulation through EGFRNet$^{mtKRAS-Lo}$, the AP-1 binding site motif for FOS/JUN dimers was enriched in the promoters of genes upregulated in mtKRAS$^{Lo}$ cells at this timepoint (data not shown). The prediction that STAT1 receives lower IF in EGFRNet$^{mtKRAS-Hi}$ is consistent with reports that mtKRAS inhibits STAT1/2 expression[49]. To directly examine STAT activity, we used luciferase reporter genes that are regulated by STAT1/2/3 TFBS (Fig. 5j). STAT1/2 reporter activity was significantly elevated in mtKRAS$^{Lo}$, while STAT3 activity was similar in mtKRAS$^{Hi}$ and mtKRAS$^{Lo}$ cells.

In summary, these data suggest that mtKRAS mediated PPIN rewiring alters signal flow through the EGFR network leading to the induction of different transcriptional programs. These

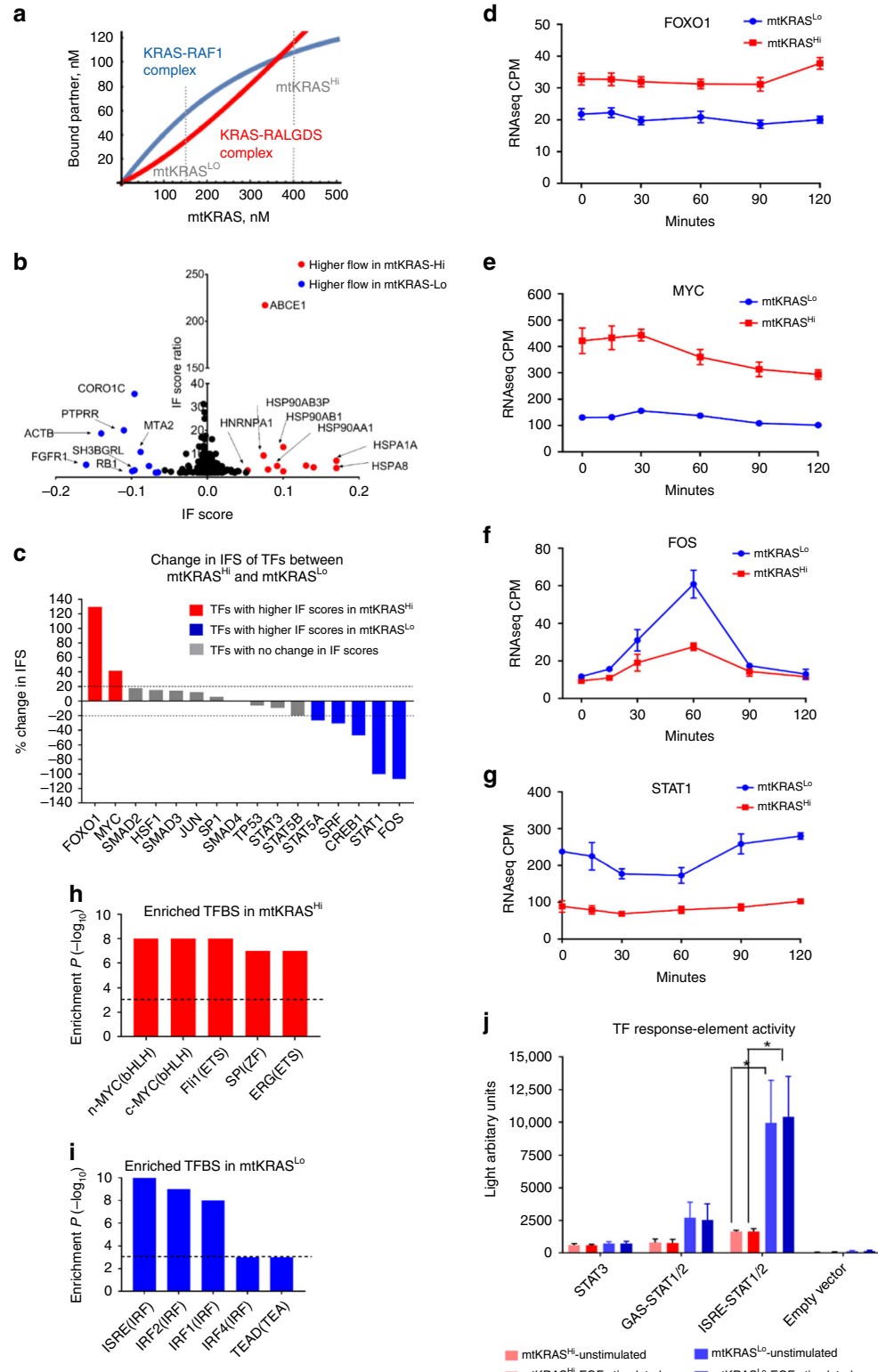

**Fig. 5 KRAS effector pathway and information flow (IF) analysis of the EGFRNet^mtKRAS-Hi and EGFRNet^mtKRAS-Lo networks. a** Dependence of KRAS-effector complex concentrations of effectors binding with high (RAF1, blue) or low affinity (RALGDS, red) on the abundance of mtKRAS. Broken gray lines indicate KRAS concentrations in mtKRAS^Lo and mtKRAS^Hi cells. **b** Plot showing nodes in the top 5th percentile in terms of their IFS and predicted to receive more flow in EGFRNet^mtKRAS-Hi (red) or EGFRNet^mtKRAS-Lo (blue). **c** Transcription factors (TFs) with at least 20% higher information flow in EGFRNet^mtKRAS-Hi (red) and EGFRNet^mtKRAS-Lo (blue). Gene expression of **d** FOXO1, **e** MYC, **f** FOS, and **g** STAT1 as determined by RNAseq analysis of TGFα stimulated cells. ***EdgeR FDR < 0.001; ****EdgeR FDR < 0.0001. The top five enriched transcription factor binding (TFBS) site motifs in the promoters of genes upregulated in **h** mtKRAS^Hi and **i** mtKRAS^Lo cells. **j** Reporter gene assays of the activity of STAT1/2 and STAT3. Error bars represent standard deviation, and P values in K were determined by a two-tailed Student's t-test. *P < 0.05; **P < 0.01. The data represent three independent experiments. Source data are provided as a Source Data file. Mathematica code for 5A is provided in Supplementary Software 1.

analyses also support our PPIN reconstruction and the functional consequences of PPIN rewiring by unbiased global approaches.

**Alterations in rewired baits predict CRC patient survival**. The results presented above suggest that PPIN rewiring is associated with mtKRAS signaling and oncogenic potential. Therefore, we investigated whether alterations in bait proteins showing the most rewiring were prognostic of CRC patients' clinical outcomes. We assessed survival data of 629 CRC patients from the TCGA dataset[50]. Fifty-four percent of patients had genetic or expression alterations in the top 20 most rewired bait proteins, as defined by the sum of rewired interactions associated with each bait (Fig. 6a, Supplementary Data 15). Patients with alterations in top 20 most rewired baits had significantly poorer survival ($P < 0.04$) than patients without alterations in these proteins (Fig. 6b). Ten-year survival was 34.61% for patients with alterations in the top 20 rewired baits vs. 61.43% for patients without. These data were robust to removing the bottom 50% of least significant (based on the significance $A$ value) rewired interactions from the rewiring analysis and recalculating the top 20 most rewired baits. 18 of the 20 original top 20 baits were the same in this analysis (data not shown). By contrast, there was no significant association between alterations in the 20 least rewired bait proteins (Fig. 6c) and patient survival ($P = 0.20$) (Fig. 6d), although all of these baits were preselected because of their roles in the EGFR pathway. This association with survival became even stronger ($P = 9.855\mathrm{e}{-3}$), if we defined the top 20 most rewired baits based on interactions that were selectively enhanced in the mtKRAS[Hi] cells (Supplementary Fig. 9A).

To assess the accuracy of the top-20 bait proteins to classify patients into high and low risk groups, we trained a Lasso classifier[51] using the CRC patient data from TCGA. Five-fold cross-validation by subsampling the patient data into training (80%) and test (20%) datasets gave an accuracy of up to 0.79 (mean 0.70) and an area under the ROC curve (AUC) of 0.763 (Supplementary Fig. 9B). A similar classification using the bottom 20 least rewired proteins gave a much lower mean accuracy of 0.4 and AUC of 0.522. Several top rewired baits were highly connected nodes. Therefore, to ascertain that the association with patient outcomes was due to the rewiring of these baits and not just because they were highly connected, we selected the bottom 36 least rewired baits that together accounted for at least the same number of interactions as the top 20 baits. Patients with alterations in the top 20 rewired baits again showed significantly poorer survival ($P < 0.017$, log-rank test) than patients with alterations in these baits. (Supplementary Fig. 9C). Patients with alterations in the top 20 rewired baits also showed significantly poorer survival after adjusting for age and tumor stage ($P < 0.03$, log-rank test). These data suggest that the proteins, which we found to be the most rewired in mtKRAS[Hi] cells, are clinically relevant as alterations in these proteins are prognostic of CRC patient outcomes.

## Discussion

Global PPIN mapping has validated the concept that the cell organizes its proteome as modules of PPIs that enable it to carry out its specific biological functions[1,2,4]. Many disease-associated mutations affect PPIs[25], but the extent of adaption to disease mutations at a PPIN level and its functional consequences are unknown. Our comparative mapping of >6000 PPIs in the EGFR network in cells with low and high mtKRAS signaling reveals a widespread rewiring of the EGFR signaling network. Interestingly, rewiring percolates through the whole network and alters interactions that occur between core components of the EGFR pathway as well as interactions between proteins involved in

downstream and seemingly peripheral processes. This suggests that enhanced mtKRAS activity results in extensive adaptive changes that are reflected by a reorganization of the PPIN. Genetic mutations associated with disease often alter PPIs[25]. However, the deep network propagation of PPI changes arising from a single mutation was unexpected and may explain why blocking mtKRAS signaling by inhibiting single effector pathways is ineffective[8]. Our global analysis and validation of the functional consequences of PPIN rewiring facilitates the rational design of combinatorial targeting of mtKRAS effectors, especially as PPIs gained in mtKRAS[Hi] cells inversely correlate with CRC patient survival. For instance, HSPs receive high IF in mtKRAS[Hi] cells, and HSP90 inhibitors recently were found to enhance the effects of conventional CRC drug therapies[52]. Likewise, our results that phosphorylation changes contribute to PPIN rewiring may "repurpose" kinase inhibitors as PPIN rewiring agents. We recently showed that mtKRAS also profoundly changes the metabolic and transcriptional landscapes of CRC cells[53] confirming that mtKRAS widely affects cellular regulation on different levels. Surprisingly, our analysis shows that a low-level expression of mtKRAS is compatible with normal (i.e., untransformed) biochemical and biological behavior. This finding suggests that oncogenic mutations must reach a threshold activity before they produce a pathogenetic phenotype. While this view challenges the genetic mutational dogma of carcinogenesis, it reconciles with recent data finding oncogenic mutations in normal tissues[54,55]. The easy accessibility and analysis of our PPIN data through PRIMESDB.eu and DyNet[56], an application for the visualization and analysis of dynamic molecular interaction networks will support systematic efforts of combinatorial mtKRAS pathway targeting.

While our study is comprehensive and integrates PPIN reconstruction with computational model analysis of network functions, it also has limitations. Although we have thoroughly characterized the two cell lines by WGS, biochemical and biological assays, we cannot formally exclude that differences other than mtKRAS activities contribute to our results, e.g., differential bait expression vs. endogenous levels, epigenetic differences, or nonsynonymous mutations that affect splicing or codon usage. The influence of such factors could be addressed by reconstitution experiments that titrate mtKRAS dosage and by studying other isogenic cell line pairs. Investigating all these aspects was beyond the scope of this study. However, we assessed and found no statistical association between rewiring and alternative splicing (data not shown), which in binary interaction screens substantially changed PPIs[57]. It also will be important to test whether the observed PPI rewiring is common to different mtKRAS cancer types. Our findings that PPI changes correlate with CRC patient prognosis and often affect proteins that are synthetic lethal with KRAS[G13D][33] indicate that a core signature of consistently altered PPIs may exist in mtKRAS cells.

In summary, these results suggest that dynamic PPIN adaptations play major roles in translating the effects of genetic mutations into quantitative functional effects that re-direct information flow through signaling networks and reprogram biological outcomes.

## Methods

**Cell lines and cell culture**. HCT116 (mtKRAS[Hi]) and HKE3 (mtKRAS[Lo]) cells[11] were provided by Drs Shirasawa and Sasazuki. Cell lines were authenticated by whole genome sequencing (Supplementary Datas 6–8) and RNAseq as recently described[12].

**Baits and expression vectors**. For AP-MS experiments 95 baits (Supplementary Data 1) were selected that provide a broad coverage of the known EGFR signaling network. Bait cDNAs were obtained from Origene and cloned into the SF-TAP

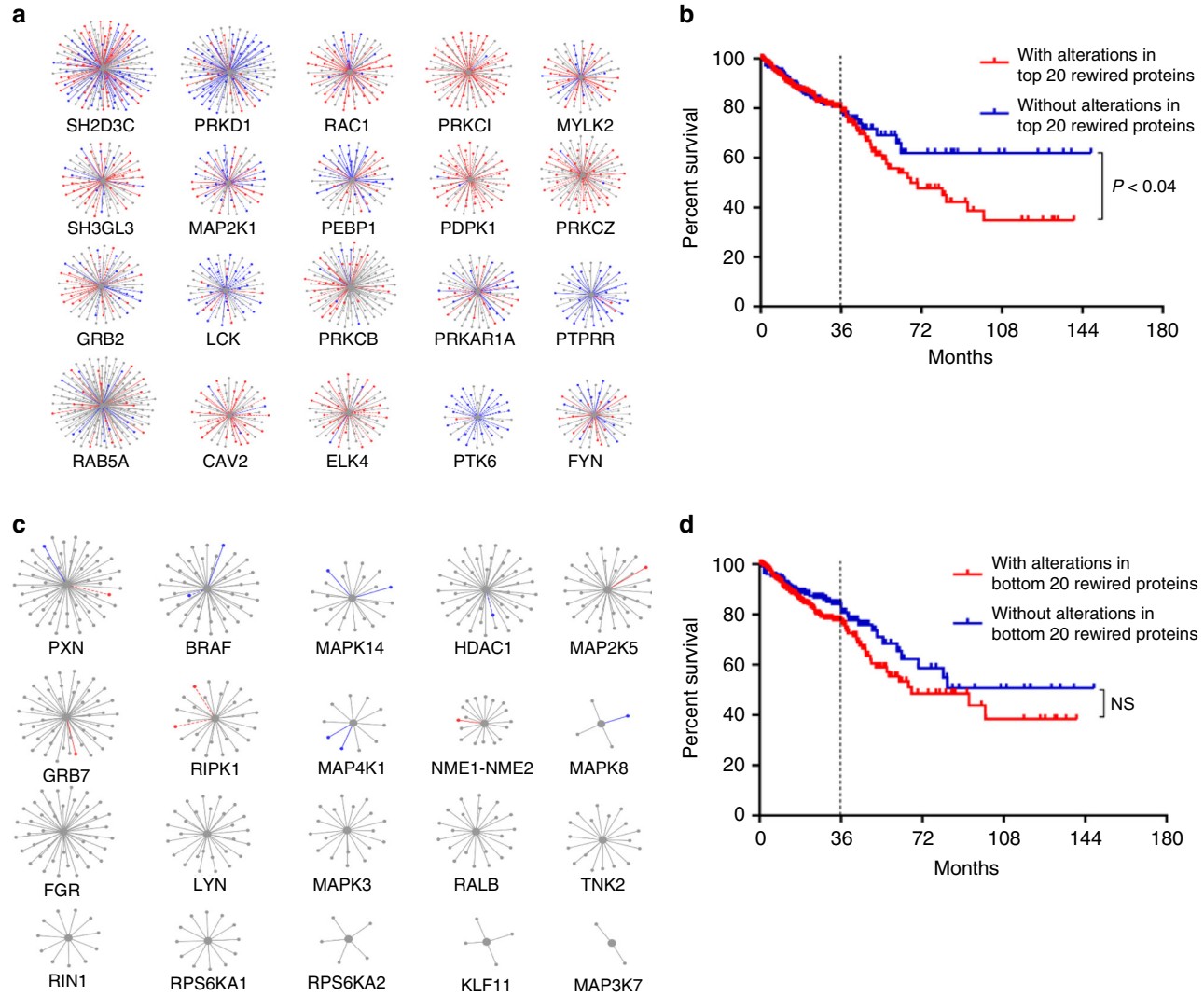

**Fig. 6 PPIN rewiring and CRC prognosis. a** The top 20 most rewired bait proteins. Interactions where the prey protein was identified only in EGFRNet^mtKRAS-Hi or EGFRNet^mtKRAS-Lo are shown as solid red or blue lines, respectively. Rewired bait-prey interactions where prey abundance was significantly higher in EGFRNet^mtKRAS-Hi or EGFRNet^mtKRAS-Lo are shown as dotted red or blue lines, respectively. Bait–prey interactions which were not significantly different are gray. **b** Six hundred and twenty-nine CRC patients from the TCGA were divided into two groups, those with alterations in the top 20 rewired bait proteins (339; 54%) and those without alterations in the top 20 rewired bait proteins (290; 46%). The alterations assessed were mutations, copy number changes, mRNA expression changes, and protein expression changes. Kaplan–Meier survival curves were plotted for the two patient groups using PRISM 7.0.3. Five-year survival was 53.5% for patients with genetic alterations affecting the top 20 rewired nodes compared to 68.5% for patients without alterations in these proteins, and ten-year survival was 34.61 vs. 61.43%, respectively. The log-rank test was used to assess statistical significance. **c** The bottom 20 least rewired bait proteins. **d** There was no significant (NS) difference in survival between patients with alterations in the bottom 20 least rewired bait proteins and those without. Source data are provided as a Source Data file.

vector[58] with the FLAG-tag at the N-terminus using the Gateway cloning system (Thermo Fisher).

**AP-MS experiments**. Cells were transfected with baits titrated to achieve similar expression levels between the cell lines and labeled with stable isotope (SILAC) medium (Fig. 1). Baits were immunoprecipitated with anti-FLAG-M2 conjugated agarose beads (Sigma-Aldrich A2220), digested with trypsin and analyzed by quantitative MS using a Q-Exactive mass spectrometer (Thermo Fisher Scientific). Data were analyzed with MaxQuant[59] and HiQuant[19] software packages. See Supplementary Methods, sections 8–12, for a detailed description.

**Protein expression profiling**. Lysates of SILAC labeled cells were digested with trypsin and analyzed on an Orbitrap Fusion Tribrid mass spectrometer (Thermo Fisher Scientific). Data were analyzed with MaxQuant[59] and HiQuant[19] software packages. See Supplementary Methods, sections 13–15, for a detailed description.

**Phosphoproteomics**. Cell lysates were digested with trypsin, phosphopeptides were enriched using TiO₂ beads and analyzed on a Q-Exactive mass spectrometer

(Thermo Fisher Scientific). MS data were analyzed with MASCOT 2.4 (Matrix Science Ltd) and Progenesis 4 (Nonlinear Dynamics). See Supplementary Methods, sections 16–18, for a detailed description.

**Construction of protein–protein interaction networks (PPIN)**. The EGFR-Net^mtKRAS-Hi and EGFRNet^mtKRAS-Lo networks were separately constructed by combining bait–prey interactions from each of the 95 chosen baits. Bait-prey interactions were included in the networks, if the abundance of the prey protein in the pull-down was significantly higher ($P \leq 0.05$) than in empty vector controls and the significance A value for the prey protein was also ≤0.05. To identify interactions that were significantly "rewired" in the HCT116 (EGFRNet^mtKRAS-Hi) network compared to the HKE3 (EGFRNet^mtKRAS-Lo) network, we used HiQuant to directly compare the SILAC data from the two cell lines. We defined interactions as being "rewired" in EGFRNet^mtKRAS-Hi, if the abundance of the prey protein in the pull-down was significantly different ($P \leq 0.05$) compared to EGFRNet^mtKRAS-Lo and the significance A value for the prey protein was also ≤0.05. Interactions that were identified in only one EGFRNet, but where prey abundance was subsequently not found to be statistically significantly different in the respective bait–prey complexes in the two cell lines were not considered as rewired interactions. The top rewired

nodes in EGFRNet$^{mtKRAS-Hi}$ were identified as those with most rewired interactions. The topological properties of the networks including node degree, betweenness centrality, clustering coefficient and network scale-freeness were analyzed using the NetworkAnalyzer application[60] in Cytoscape 3[61]. Cytoscape session files for the EGFRNet$^{mtKRAS-Hi}$, EGFRNet$^{mtKRAS-Lo}$ networks can be provided upon request.

**Protein abundance and phosphorylation enrichment analysis**. To investigate whether nodes in the EGFRNet$^{mtKRAS-Hi}$ network were enriched for differentially abundant proteins, a hypergeometric test was performed with the following parameters:

$$p(X \geq k) = \sum_{x=k}^{n} \frac{\binom{K}{x}\binom{N-K}{n-x}}{\binom{N}{n}},$$

where $N$ is the total number of proteins assayed in the protein expression analysis, $n$ the total number of differentially abundant proteins identified, $K$ the number of proteins in the EGFRNet$^{mtKRAS-Hi}$ network that were assayed in the protein expression analysis, $k$ the number of differentially abundant proteins observed in the EGFRNet$^{mtKRAS-Hi}$ network.

A similar analysis was conducted to determine whether rewired nodes were enriched for differentially abundant or phosphorylated proteins. See Supplementary Methods, section 19, for a detailed description.

**Equilibrium binding model of RAS binding partners to RAS-GTP**. In order to determine how the concentrations of KRAS-effector complexes change with the concentration of active RAS in mtKRAS$^{Hi}$ and mtKRAS$^{Lo}$ cells, we developed a dynamic mathematical model that allowed us to investigate how competition for the single effector binding site on RAS and different abundances of low and high affinity RAS effectors impact the formation of RAS-effector complexes. See Supplementary Methods, sections 22–25, for a detailed description.

**Information flow analysis of EGFRNet$^{mtKRAS-Hi}$ and EGFRNet$^{mtKRAS-Lo}$ networks**. In order to analyze how the EGFR PPINs transduce information we employed a computational modeling approach called information flow (IF) analysis[62,63]. To perform IF analysis from the EGFR at the cell membrane to nuclear transcription factors (TFs), the two EGFRNets were first supplemented with publicly available prey–prey interactions from InnateDB[64] and 122 additional nodes that are known to be involved in EGFR signaling[18] but were not chosen as bait proteins in our AP-MS experiments (Supplementary Data 14). These networks are referred to as the HCT116$^{IFANET}$ and HKE3$^{IFANET}$. Information flow analysis was implemented using the CytoITMprobe software (damping factor = 0.85; channel model selected)[41], selecting EGFR as the source node of signaling and 19 downstream TFs (Supplementary Data 14) as the sinks for the information flow. Information flow scores were determined by measuring how much information flows through each node in the HCT116$^{IFANET}$ and HKE3$^{IFANET}$ networks. See Supplementary Methods, section 26, for a detailed description.

**Gene ontology, pathway, and transcription factor binding site analyses**. Gene ontology (GO) and pathway analyses were performed using InnateDB.com[64] regarding GO terms or pathways with an FDR < 0.05 as significantly enriched. Transcription factor binding site analysis was undertaken using the findMotifs.pl program in HOMER v4.8[65], with the human hg38 promoter set in order to identify enriched motifs. See Supplementary Methods, section 27, for a detailed description.

**Analysis of CRC patient data**. Survival data of 629 CRC patients were obtained from the TCGA[50], and correlated with alterations in genes encoding either the top 20 most rewired or the bottom 20 least rewired bait proteins. The alterations included were mutations, copy number changes, mRNA expression changes, and protein expression changes. As an additional control we selected a set of 36 baits that accounted for the same number of interactions as the top 20 baits in the network. The Kaplan–Meier curves were plotted using PRISM 7.0.3. See Supplementary Methods, section 27, for a detailed description.

**Western blotting**. Cells were lysed in 1% NP40, 20 mM Tris-HCl pH 7.5, 150 mM NaCl, 1 mM MgCl$_2$) supplemented with protease inhibitor cocktail (Roche) and phosphatase inhibitors (2 mM sodium orthovanadate, 10 mM sodium fluoride and 10 mM β-glycerophosphate; all from Sigma-Aldrich) for 10 min at 4 °C. Lysates were cleared by centrifugation at 20,000 × $g$ for 10 min, and adjusted to equal protein concentrations. Proteins were separated by sodium-dodecylsulfate poly-acrylamide gel electrophoresis (SDS-PAGE) and transferred to polyvinylidene difluoride (PVDF) membranes. Blots were incubated with the respective antibodies and developed using Enhanced Chemiluminescence (ECL; Thermo Fisher) according to the manufacturer's instructions. Blots were quantified using the Image J software and phospho-specific antibody signals were normalized to the total abundance of the respective proteins.

**Luciferase assays**. The transcription factor response element activity was assessed with luciferase constructs bearing response elements for STAT1 (4xGAS response element; Stratagene, #219091–51), STAT1/STAT2 (IRSE/interferon alpha response element[66]) and STAT3 (4xm67 response element[67]). A CMV-β-gal plasmid was co-transfected as a control of transfection efficiency. Forty-eight hours post transfection cells were stimulated with 10 nm human EGF (Roche; #11376454001) for 5 h before luciferase and β-gal activity were measured using luciferase assay (Promega, #E4030) and β-galactosidase assay kits (Promega, #E2000). Luciferase activity was normalized against β-gal activity to correct for transfection efficiency.

**Transcriptional profiling**. HCT116 (mtKRAS$^{Hi}$) and HKE3 (mtKRAS$^{Lo}$) cells were serum starved for 18 h before stimulation with TGF-α (0.01 µg/ml, Abcam) for 0, 15, 30, 60, 90, and 120 min. RNA was extracted using the TRIzol reagent (Thermo Fisher Scientific) from three biological replicates at each time point. RNAseq was performed with an Illumina HiSeq 2500 machine using a v2 High Output 100 cycle Kit (1 × 100 bp SR). Data were processed as described in the Supplementary Methods.

**Reporting summary**. Further information on research design is available in the Nature Research Reporting Summary linked to this article.

## Data availability
RNAseq data were deposited in the Gene Expression Omnibus under accession number GSE105094. These data were used in Fig. 5d–i and Supplementary Fig. 5B. Whole genome sequencing data were deposited in the NCBI short read archive under accession number PRJNA374513. They were used in Supplementary Fig. 5A. Proteomics data were deposited in the PRIDE database under the following accession numbers: PXD016512, PXD016505, PXD016465, PXD016464, PXD016463, PXD016462, PXD016461 for the AP-MS data, PXD016549 for the protein expression profiling data, and PXD016431 for the phosphoproteomics data. AP-MS data can be visualized and browsed in PRIMESDB, a database developed for this project and described in detail in the Supplementary Data. PRIMESDB is accessible at primesdb.eu. is an observer member of The International Molecular Exchange (IMEx) consortium, the international standards body for the curation and exchange of published protein-protein interaction data[68]. These data were used in Figs. 2, 3, 5b, 6 and Supplementary Figs. 2, 4, 6, 7, 9. All PPI data generated in this study also been deposited with IMEx (IMEx accession number IM-26434). TCGA data were obtained from https://www.cbioportal.org/study/summary?id=coadread_tcga. The source data underlying Figs. 2a–c, 3a–c, 4a–d, 5a–j, 6a–d and Supplementary Figs. 1b–i, 2a–i, 3a–c, 4, 5a–f, 6a, b, 7, 8b–e, 9a–c are provided as a Source Data file

## Code availability
Mathematica, R, and Cytoscape files with code are provided as Supplementary Software: Supplementary Software 1. Mathematica code for Fig. 5a. Supplementary Software 2. Cytoscape session file for Supplementary Fig. 2A. Supplementary Software 3. R-code and source data used for Supplementary Fig. 3A. Supplementary Software 4. Cytoscape session file for Supplementary Fig. 5C, D. Supplementary Software 5. Cytoscape session file for Supplementary Fig. 5E, F. Supplementary Software 6. Cytoscape session file for Supplementary Fig. 6A, B. Supplementary Software 7. R-script for Supplementary Fig. 9B.

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

## Acknowledgements

This work was supported by European Union FP7 Grant No. 278568 "PRIMES" and Science Foundation Ireland Investigator Program Grant 14/IA/2395 to W.K. B.K. is supported by SmartNanoTox (Grant no. 686098), NanoCommons (Grant no. 731032), O.R. by MSCA-IF-2016 SAMNets (Grant no. 750688). D.M. is supported by Science Foundation Ireland Career Development award 15-CDA-3495. I.J. is supported by the Canada Research Chair Program (CRC #225404), Krembil Foundation, Ontario Research Fund (GL2-01-030 and #34876), Natural Sciences Research Council (NSERC #203475), Canada Foundation for Innovation (CFI #225404, #30865), and IBM. D.J.L. is supported by EMBL Australia. O.S. is supported by ERC investigator Award ColonCan 311301 and CRUK. I.S. is supported by the Canadian Cancer Society Research Institute (#703889), Genome Canada via Ontario Genomics (#9427 & #9428), Ontario Research fund (ORF/

DIG-501411 & RE08-009), Consortium Québécois sur la Découverte du Médicament (CQDM Quantum Leap) & Brain Canada (Quantum Leap), and CQDM Explore and OCE (#23929). T.C. was supported by a Teagasc Walsh Fellowship. MU and KB are supported by the Tistou & Charlotte Kerstan Stiftung. We thank Prof M. Uhlen for discussions and critical review of the HKE3 and HCT116 genome analysis. PRIMESDB (primesdb.eu) is supported by use of the NeCTAR Research Cloud and by eResearch SA. The NeCTAR Research Cloud is a collaborative Australian research platform supported by the National Collaborative Research Infrastructure Strategy.

## Author contributions

The study was designed by W.K., D.J.L., and K.Bo. with input from M.U., L.S., C.K., and I.S. M.A.J., S.K., C.R., N.H., F.K., T.L., L.I., K.Bo., and V.W. performed proteomics experiments. S.K., C.H.M., A.K., D.M., M.A.J., L.D., N.R., C.K., R.P., P.C., and O.S. performed biochemical and biological experiments. E.F. and C.A.S. did the whole gen-ome sequencing and some of the RNAseq experiments. D.J.L., S.Sr., M.A.J., K.Br., S.Sh., K.Bo., J.B., and M.A. analyzed the data. T.C. performed the information flow analysis and corresponding experiments including RNAseq with M.A.L., A.K., and D.J.L. PRIMESDB was developed by M.B.L., K.Br., P.P., and D.J.L. I.J., M.K., J.C., and I.S. contributed the MYTH experiments. B.N.K. and O.R. generated the mathematical model. LFIM and C.R. performed the dynamic complex analysis. D.J.L., S.Sr., and W.K. wrote the paper with contributions from all other authors.

## Competing interests

The authors declare no competing interests.

## Additional information

Susan A. Kennedy[1,27], Mohamed-Ali Jarboui[2,3,27], Sriganesh Srihari[4,5,27], Cinzia Raso[1,27], Kenneth Bryan[4], Layal Dernayka[2], Theodosia Charitou[1,4], Manuel Bernal-Llinares[4], Carlos Herrera-Montavez[1], Aleksandar Krstic[1], David Matallanas[1], Max Kotlyar[6], Igor Jurisica[6,7,8], Jasna Curak[9,10,11], Victoria Wong[9,10,11], Igor Stagljar[9,10,11,12], Thierry LeBihan[13], Lisa Imrie[13], Priyanka Pillai[4], Miriam A. Lynn[4], Erik Fasterius[14], Cristina Al-Khalili Szigyarto[14,15], James Breen[16,17], Christina Kiel[1,18,19], Luis Serrano[18], Nora Rauch[1], Oleksii Rukhlenko[1], Boris N. Kholodenko[1,19,20], Luis F. Iglesias-Martinez[1], Colm J. Ryan[1,21], Ruth Pilkington[1], Patrizia Cammareri[22], Owen Sansom[22,23], Steven Shave[24], Manfred Auer[24], Nicola Horn[2], Franziska Klose[2], Marius Ueffing[2], Karsten Boldt[2,28]*, David J. Lynn[4,25,28]* & Walter Kolch[1,19,26,28]*

[1]Systems Biology Ireland, University College Dublin, Dublin, Ireland. [2]Institute for Ophthalmic Research, University of Tübingen, Tübingen, Germany. [3]Werner Siemens Imaging Center, University of Tübingen, Tübingen, Germany. [4]EMBL Australia Group, South Australian Health and Medical Research Institute, Adelaide, SA 5000, Australia. [5]QIMR-Berghofer Medical Research Institute, Brisbane, QLD 4006, Australia. [6]Krembil Research Institute, University Health Network, Toronto, Canada. [7]Departments of Medical Biophysics and Computer Science, University of Toronto, Toronto, Canada. [8]Institute of Neuroimmunology, Slovak Academy of Sciences, Bratislava, Slovak Republic. [9]Donnelly Centre, University of Toronto, Toronto, Canada. [10]Department of Biochemistry, University of Toronto, Toronto, Canada. [11]Department of Molecular Genetics, University of Toronto, Toronto, Canada. [12]Mediterranean Institute for Life Sciences, Split, Croatia. [13]Synthetic and Systems Biology, University of Edinburgh, Edinburgh, UK. [14]School of Biotechnology, KTH Royal Institute of Technology, Stockholm, Sweden. [15]Science for Life Laboratory, KTH Royal Institute of Technology, Stockholm, Sweden. [16]School of Biological Sciences, University of Adelaide Bioinformatics Hub, Adelaide, SA, Australia. [17]Computational & Systems Biology Program, South Australian Health and Medical Research Institute, Adelaide, SA, Australia. [18]Centre for Genomic Regulation, Barcelona Institute of Science and Technology, Barcelona, Spain. [19]Conway Institute, University College Dublin, Dublin, Ireland. [20]Department of Pharmacology, Yale University School of Medicine, New Haven, CT, USA. [21]School of Computer Science, University College Dublin, Dublin, Ireland. [22]Cancer Research UK Beatson Institute, Glasgow, UK. [23]Institute of Cancer Studies, Glasgow University, Glasgow, UK. [24]School of Biological Sciences and School of Biomedical Sciences, University of Edinburgh, Edinburgh, UK. [25]College of Medicine and Public Health, Flinders University, Bedford Park, SA 5042, Australia. [26]School of Medicine, University College Dublin, Dublin, Ireland. [27]These authors contributed equally: Susan A. Kennedy, Mohamed-Ali Jarboui, Sriganesh Srihari, Cinzia Raso. [28]These authors jointly supervised: Karsten Boldt, David J. Lynn, Walter Kolch. *email: karsten.boldt@uni-tuebingen.de; david.lynn@sahmri.com; walter.kolch@ucd.ie

