## [Peer Review File · Nature Communications]

Reviewers' comments:

Reviewer #1 (Remarks to the Author): expert in signalling

The manuscript "Extensive rewiring of the epidermal growth factor receptor protein-protein interaction and signaling network in colorectal cancer cells expressing transforming levels of oncogenic KRAS G13D" describes an extensive and thorough characterization of a pair of (mostly) isogenic cells. The authors list includes established and outstanding experimental, computational, and systems biologists with true expertise in the Ras pathway. Additionally, the work investigates how oncogenic KRAS promotes cancer by means of how signals propagate downstream from it, which is a long-standing, well-studied, and still incompletely understood problem.

However, there are some significant flaws to this study that limit the extent to which conclusions can be drawn.

The experimental design involves comparing a pair of isogenic cells. That is a clever, and increasingly common, approach for studying how a single mutant protein will disrupt signaling. The authors chose HCT116 colon cancer cells, which carry a KRAS G13D mutation. For the isogenic partner, they utilized the HKE3 cell line derived many years ago (Shirasawa et al, Science, 1993). It seems that the original interpretation of HKE3 cells was that the KRAS G13D allele of HCT116 was replaced with a KRAS WT allele to create "HKE3". However, in a very important and thorough characterization by the authors, the authors find that KRAS G13D is still present in HKE3 cells. Additionally, the authors demonstrate that they do not have a polyclonal mix of HKE3 and HCT116, but that there is a residual level of KRAS G13D in HKE3.

On the positive side, the HKE3 cells are (by 1993 manuscript) non-transforming relative to HCT116 (by growth in soft-agar and in mice). So this is a pair of relatively isogenic cells with different phenotypes that one could study. However, the major problem is that there are likely other differences between the cells. The KRAS mutant variable is simply not as well controlled as the experimental design may at first imply.

The authors address this, in part, by looking for other single-nucleotide variants. Through computational inference, they conclude only one is likely to be functional, and they argue against. However, computational inference remains far from perfect. The authors only present a single variant in Table S6, although their phrasing in the text implies that they found additional variants. All coding variants should be listed, regardless of what the predicted effect is. That way, the study could be reassessed in the future as variant prediction continues to improve.

Additionally, there may be epigenetic changes and copy number changes that could vary between these cells. This seems likely considering that the KRAS allele has a copy number change. Array CGH would be a relatively cheap way to compare copy number status between a pair of clonal cell lines and should be performed.

That there are differences between HKE3 and HCT116, in hindsight, perhaps unsurprising. The HCT116 cell is likely partially dependent (if not truly oncogene addicted) to KRAS, so the elimination of the KRAS G13D allele may only be possible when there is one or more mutations that enable the cell to be viable after KRAS G13D loss. Additionally, the selection of clones takes the cell line through a bottleneck that may bring along additional mutations - which may or may not contribute to fitness, to KRAS signaling, and/or to PPIN measurements.

The PPIN mapping and analysis appears to be well-done and in accordance with the standards of the field. However, it was difficult for this reader to dig in too deeply or to be too interested in the findings due to the issue between HKE3 and HCT116. The language used to describe conclusions read entirely too strong considering the unexpected differences in KRAS mutant status, other changes, and likely additional changes not measured, etc., in HKE3.

The computational modeling does not seem to add a tremendous amount of value to the manuscript. The idea that a high affinity protein will bind before a low affinity protein is self-evident, and the idea that if the high affinity protein is limiting that the low affinity protein may later have more total bound (e.g. if low-affinity is much more abundant than high affinity protein) is not novel. It isn't clear that this concept is needed or necessary to explain the author's data. Moreover, it is not clear if this concept is needed for the authors claim of perfect correlation between model predictions and PPIN observations in Fig 5B. If there is something significant here, the authors should explain it more thoroughly and more clearly. If there is not something here, the authors may wish to cut it from the manuscript.

Smaller, specific, comments:

- a) Can the Genomic data be used to determine precisely where the three claimed KRAS alleles in HKE3 are located? It would be valuable to better define the KRAS genomic situation.
- b) The language of this paper was too strong. For example: page 4 "... inhibiting downstream effectors has proven ineffective due to complex feedback structures... and to the large number of effector pathways." The term "Proven" is quite strong. Perhaps, "... inhibiting downstream effectors has to date been ineffective, likely because of ..."
- c) More problematic is the use of overly strong and definitive language to describe the data presented. Due to the problematic nature of HCT116 and HKE3, the authors need to be much more cautious with their interpretations and conclusions.
- d) On pg 7 – the authors state their findings suggest many PPIs may be highly dependent on the cellular context. If true (and it likely is true), that would also suggest these results may not apply to other cancer cells, which have very, very, different sets of genomic and epigenetic alterations.
- e) Table S6. The authors state "only one expressed gene (NAV1) exhibited a non-synonymous variant predicted to alter protein function (Table S6)" The version of Table S6 that I have access to only lists one gene, NAV1. Were there non-synonymous variants NOT predicted to alter protein function? They should be listed as well. It should be more clear in the table which method(s) were used to predict function, and what those algorithms predicted. Multiple approaches should be used to support the claim that only one needs to be considered. Whether other changes (copy number changes, splice-site changes, silent mutations that may shift codon usage to a more or less common tRNA, etc.) should be discussed, too. There is much more to genomics and cancer genomics than SNVs.
- f) TCGA analysis for survival. It is unclear if multiple hypothesis testing was properly controlled for. That is, did the authors originally say "we will use exactly the top 20 genes, look at only 5 yr survival, and determine a p-value." Or did they consider multiple timepoints and multiple lists of genes of varying lengths.
- g) Mathematical model. The version of the supplementary materials available to this reviewer had the top and bottom of the equations cutoff. A reformatted version would be needed to evaluate their model.
- h) Figure 5B seems to be important, but it is not clear what is being presented. Are these predictions based on both abundances and Kd values? From what source did the Kd and abundance values come? The values used should be presented in a table. There is no consensus in the literature what the values are (e.g. wide ranges of Kd values can be found in the literature for specific interactions). Are the predictions robust to the variation that is in the literature? How robust is the "perfect correlation"? How much of the correlation is only due to Kd, how much requires abundance? Etc.
- i) The Discussion feels a bit generic re: the value of defining the PPINs. Does this study help move the field forward? How so? Are data & findings incremental or are they significant? If significant, please explain how so. Did this work uncover any important problems that should be fixed in future studies? A discussion of whether the differences between HKE3 and HCT116 were known before the study, or were discovered after the fact, could be valuable and help the field move forward and/or better anticipate similar problems.

Major suggestions:

Overall, this manuscript reads as an impressively large and thorough body of work. It seems like almost every methodology to study the data was attempted. However, the here discovered and reported differences between HKE3 and HCT116 make it much less well-suited for PPIN comparison than the pair would have seemed before this new information became available.

It is not clear how this can be reasonably addressed. An "add back" of KRAS G13D into HKE3 cells could be utilized to test whether interactions specific to HCT116 or HKE3 are flipped with the reintroduction of G13D. However, as RAS activation is not either/or but is graded with thresholds (per the authors text), this experiment comes with numerous challenges. Alternatively, Shirasawa et al appear to have developed other isogenic versions of HCT116 that did not have KRAS G13D. If multiple G13D k/o cells have similar PPINs, that would support that what is being observed is truly due to the decrease in KRAS G13D expression and not due to other, random, unshared, events. Lastly, cancer synthetic lethal experiments, often utilize sets of KRAS mutant and KRAS WT cells to find the changes that are "universal" for Mutant and WT cells and to compare differences. Validation in additional KRAS mutant/WT isogenic (or mostly) isogenic pairs could be valuable. Of course, these 3 sets of experiments are likely cost and/or time prohibitive. They are discussed simply to put the flaws of this study in context and to contribute to discussion of how to design better studies going forward.

The most important changes, in my opinion, are clear explanations of the limitations of the study and the use of much less strong language to describe the findings.

The other issue with this manuscript is it is not clear what new understanding of biology comes from the manuscript. That there are differences and "rewiring", etc., seems logical and consistent with many other studies, even if this exact study was not previously done.

Reviewer #2 (Remarks to the Author): Expert in proteomics

The manuscript by Kennedy and co-workers describes a systematic analysis of protein-protein interactions (PPIs) in two colorectal cancer cell lines that were designed to differ in the levels of oncogenic KRAS expression. To this end, the authors perform quantitative affinity purification mass spectrometry experiments. They observe extensive rewiring of the PPI network, including nodes that are far away from the immediate impact of KRAS. These perturbations appear to be relevant for CRC since they correlate with survival.

This paper addresses an exciting and highly relevant biomedical question and presents an impressive amount of data. The proteomic data appears to be of high quality and the described data analysis pipeline is sound. The modeling part also sounds convincing, although I am not able to evaluate the details of this. Overall, I think this paper will become suitable for publication when the authors have addressed a number of important points:

1. Page 6: "The baits were expressed as FLAG-tagged proteins carefully titrating transfection to achieve a modest and similar level of overexpression in both cell lines." The endogenous expression levels of the bait proteins is different. Therefore, titrating transfections to achieve "modest and similar overexpression" requires experiments such as western blots with antibodies against the bait protein (not the FLAG tag) to compare the level of exogenous and endogenous protein. Otherwise, the degree of overexpression will be modest for more abundant proteins but very high for less abundant proteins. I cannot find such western blot data. Either such data should be provided or the statement has to be reworded to indicate that the degree of overexpression is not always modest.
2. Related to the point above, it should be mentioned that network rewiring could also occur as a

result of differential bait expression. In fact, this is exactly what the authors observe for KRAS. Since this study used exogenous baits that were artificially overexpressed at a similar level in both cell lines, this aspect of network rewiring cannot be addressed.

3. While the authors validate their AP-MS data, I don't see validation for the membrane yeast two-hybrid data (MYTH) that they also report. The overall question is how sensitive and specific the MYTH data is. More generally, it is not clear to me if/what the MYTH data really contributes to this paper, especially since it cannot reveal rewiring of PPIs in response to oncogenic KRAS. Is it really worth to include this data in the paper at all?

4. Page 10: The authors find a significant correlation between between differential node abundance and network rewiring. From this, they conclude that "differences in protein expression between mtKRASHi and mtKRASLo cells can explain some of the rewired interactions.". I agree with this conclusion. In addition, the authors should also consider the alternative possibility that rewiring the interactions causes differences in protein abundance: A number of studies have shown that complex formation can stabilise proteins and thus increase steady-state protein levels (<https://doi.org/10.1016/j.cell.2016.09.015>; <https://doi.org/10.15252/msb.20177548>; <https://doi.org/10.1074/mcp.M116.060301>). This alternative possibility should at least be mentioned. The authors may also want to include a more detailed analysis using their mRNA data for both cell lines: If differential node abundance is only seen at the protein but not at the mRNA level, this suggests that differential interaction causes differences in node abundance (and not vice versa).

5. Figure 5C: According to the text this is a volcano plot. I don't fully understand this: A typical volcano plot displays the $-\log$ significance on the y axis. This plot shows the IF score ratio instead. Not unexpectedly, this ratio tends to be more extreme in the region where the IF score is close to 0 (i.e. data points around the y axis). Thus, some IF score ratios are huge although the absolute IF score is very small, which is why the proteins in this region are probably not very informative. In a typical volcano plot this region would be depopulated (proteins which do not change are also not significant).

6. Impact of phosphorylation on PPIs: Several interactions downstream of EGFR are well-known to be regulated by tyrosine phosphorylation via SH2 domains. I am missing a more detailed analysis of this: Do the authors see a correlation between differential tyrosine phosphorylation and corresponding PPIs?

7. The authors mention a number of different mechanisms that lead to network rewiring. It would be great to have an estimate how important each of these mechanisms is. Can this somehow be estimated based on the data?

8. Supplemental material Table A: The last line of the previous paragraph is inserted between the table and the legend. This should be fixed.

9. Supplemental material: The authors discuss the benefits of SILAC label swapping/switching. I agree with them that label swapping is useful, but I disagree with the statement that "label switching is not commonly done in SILAC experiments". It is actually done routinely in many labs, so this idea is not new. The authors should also stick to one term. I would prefer label swapping over label switching - this is also the term that is typically used in the proteomics community.

10. Supplemental Material: The authors write that they used a "high-resolution pre-scan" with two microscans per spectrum. I don't understand what this means. Orbitrap instruments typically use prescans for automatic gain control (AGC), but I am not aware of an option to use two such prescans. Also, it appears that no MS1 scans were made. I assume that the authors did not use prescans but full (MS1) scans at $R=30,000$? Is this indeed the case? If yes, why did they decide to record two microscans? A single full scan at a resolution of $R=60,000$ would have given better data (higher resolution) with the same duty cycle.

11. While I am not able to check all details of the "HiQuant" method they used, the data analysis strategy is sound and well described. I only have one question here: Why did the authors use "significance A" rather than "significance B" as their second filter criterion? Significance B has the additional advantage that protein abundance is taken into account. This may be useful, since SILAC ratios of lower abundant proteins tend to show a higher variability. In other words, significance A is expected to favour low abundant proteins and to bias against high abundant proteins in the pull-downs.

This review is from Matthias Selbach

Reviewer #3 (Remarks to the Author): Expert in colorectal cancer

Susan Kennedy, Walter Kolch and colleagues have used quantitative mass spectrometry (qMS) to map KRAS specific changes in protein-protein interactions in two isogenic CRC cell lines that express either high or low levels of mutated KRAS. They found interesting rewiring of the signalling network as an effect following mutations in KRAS. Even though the study has many strengths, it has also some major shortcomings.

The authors have based their study on colorectal cancer (CRC) cells expressing transforming (high) levels of KRASG13D (mtKRAS). Unfortunately, the cells HCT116 and MKE are both of MSI (microsatellite instable) phenotype. Those cancer cells are known to have specific characteristics compared to non-MSI tumors. MSI tumors stands for only about 15% of CRCs and can therefore reflect colorectal cancer on general terms. The specific molecular nature of MSI tumors might have had impact on the given results.

Even more problematic, HCT116 and MKE harbour a not only a KRAS G13D mutation but also a PIK3CA mutation, protein involved in the same signalling network. This can of course affect the cellular response and specific protein-protein interactions. One way of put light on this problem would be to include more cell lines with different molecular characteristics, but this is of course probably a very different paper in the end.

The survival analyses using the TCGA data set were analysed without taken other molecular and clinical data into account. Why were not MSI cases analysed separately? This would be specifically important taken their cell lines with MSI status into account. Is the mild prognostic effect seen in both MSI and non-MSI cases?

How is stage distributed among the two groups in the survival analyses? I would suggest to also include a multivariable Cox regression survival model in order to understand the prognostic information better. It can be somewhat misleading to only include univariable analyses.

Moreover, G13D is stated in the introduction to be the second most common specific KRAS mutation in CRC, and reference no. 7 is given. I would suggest the authors to carefully check this statement.

Point-by-point response to the referees' comments

(For ease of reading the reviewers' comments are in black and our replies are in blue.)

We would like to thank the reviewers for careful reading and important comments which we have tried to address as comprehensively as possible in the revised manuscript.

Reviewers' comments:

Reviewer #1 (Remarks to the Author): expert in signalling

The manuscript "Extensive rewiring of the epidermal growth factor receptor protein-protein interaction and signaling network in colorectal cancer cells expressing transforming levels of oncogenic KRAS G13D" describes an extensive and thorough characterization of a pair of (mostly) isogenic cells. The authors list includes established and outstanding experimental, computational, and systems biologists with true expertise in the Ras pathway. Additionally, the work investigates how oncogenic KRAS promotes cancer by means of how signals propagate downstream from it, which is a long-standing, well-studied, and still incompletely understood problem.

We appreciate this very positive overall appraisal which nicely summarises the motivation and aims of our work.

However, there are some significant flaws to this study that limit the extent to which conclusions can be drawn.

The experimental design involves comparing a pair of isogenic cells. That is a clever, and increasingly common, approach for studying how a single mutant protein will disrupt signaling. The authors chose HCT116 colon cancer cells, which carry a KRAS G13D mutation. For the isogenic partner, they utilized the HKE3 cell line derived many years ago (Shirasawa et al, Science, 1993). It seems that the original interpretation of HKE3 cells was that the KRAS G13D allele of HCT116 was replaced with a KRAS WT allele to create "HKE3". However, in a very important and thorough characterization by the authors, the authors find that KRAS G13D is still present in HKE3 cells. Additionally, the authors demonstrate that they do not have a polyclonal mix of HKE3 and HCT116, but that there is a residual level of KRAS G13D in HKE3.

This is correct, and we have performed a thorough genetic, biochemical and biological characterisation of this residual KRAS G13D, which is shown in Fig. S1. In addition, in the revised paper we also have included further genetic analysis as suggested by this reviewer, adding 3 new Supplementary Tables (Tables S6-8).

On the positive side, the HKE3 cells are (by 1993 manuscript) non-transforming relative to HCT116 (by growth in soft-agar and in mice).

We have repeated the soft agar experiments as reported by Shirasawa et al. (Fig. S1G) as well as other biochemical and biological experiments (Fig. S1C-H) to confirm that the HKE3 cells are not transformed.

So this is a pair of relatively isogenic cells with different phenotypes that one could study. However, the major problem is that there are likely other differences between the cells. The KRAS mutant variable is simply not as well controlled as the experimental design may at first imply.

The authors address this, in part, by looking for other single-nucleotide variants.

We performed whole genome sequencing (WGS) to assess differences between the two cell-lines. The WGS data are of high quality (30x genome coverage) and cover >99% of the genome of both the HCT116 and HKE3 cell lines as shown in Table B of the Supplemental Methods.

Through computational inference, they conclude only one is likely to be functional, and they argue against. However, computational inference remains far from perfect.

We concede that computational inference is not perfect, but currently there is no alternative method for the analysis of genome sequences and variants. However, to ensure that our analysis is state of the art, we completely re-analysed our WGS data using the Broad Institute's Genome Analysis Toolkit (GATK) pipelines, which have been employed in many other papers analysing genome sequences. This pipeline is less stringent than the one we had previously used, and the results are reported in 3 new Supplementary Tables (Tables S6-S8). To predict the impact of SNVs we used the Ensembl Variant Effect Predictor ¹. Although using this pipeline we detect more SNVs, the main conclusion holds up, i.e. that the contribution of SNVs to PPIN rewiring is minimal.

The authors only present a single variant in Table S6, although their phrasing in the text implies that they found additional variants. All coding variants should be listed, regardless of what the predicted effect is. That way, the study could be reassessed in the future as variant prediction continues to improve.

We have completely revised Table S6 to now include all synonymous and non-synonymous variants differing between HCT116 and HKE3 cell lines. In addition, we have included a description of the methods and a summary of the main findings in the Supplementary Materials p.25/26.

Additionally, there may be epigenetic changes and copy number changes that could vary between these cells. This seems likely considering that the KRAS allele has a copy number change. Array CGH would be a relatively cheap way to compare copy number status between a pair of clonal cell lines and should be performed.

As requested, we have analysed copy number variations using our HCT116 and HKE3 WGS data. The results are included in the Supplemental data as Fig. S5A and Table S8. In addition, we also list structural variants in Table S7.

That there are differences between HKE3 and HCT116, in hindsight, perhaps unsurprising. The HCT116 cell is likely partially dependent (if not truly oncogene addicted) to KRAS, so the elimination of the KRAS G13D allele may only be possible when there is one or more mutations that enable the cell to be viable after KRAS G13D loss.

Additionally, the selection of clones takes the cell line through a bottleneck that may bring along additional mutations - which may or may not contribute to fitness, to KRAS signaling, and/or to PPIN measurements.

The HCT116 cell have a higher basal apoptosis rate than HKE3 cells (as shown in Fig. 4B), which is due to the pro-apoptotic activity of mutant KRAS as we have shown before ². In this previous paper, we also showed that the wildtype KRAS antagonizes the pro-apoptotic activity of mutant KRAS both *in vitro* as well as in human colorectal cancers. In fact, the selective depletion of the wildtype KRAS allele tripled the apoptosis rate of HCT116 cells. These previous results suggest that HCT116 are not addicted to mutant KRAS for survival, and that compensatory mutations are not needed. The avoidance of such selection pressure was a reason why we chose HCT116 – HKE3 cell line pair. We have included a short statement explaining this rationale into the results section: "Furthermore, HCT116 are not addicted to mtKRAS for survival, minimizing selection pressure to acquire compensatory mutations when mtKRAS dosage is reduced ²."

The PPIN mapping and analysis appears to be well-done and in accordance with the standards of the field. However, it was difficult for this reader to dig in too deeply or to be too interested in the findings due to the issue between HKE3 and HCT116. The language used to describe conclusions read entirely too strong considering the unexpected differences in KRAS mutant status, other changes, and likely additional changes not measured, etc., in HKE3.

We have carefully gone through the whole manuscript to assure that conclusions align with supporting evidence and that overinterpretation is avoided.

The computational modeling does not seem to add a tremendous amount of value to the manuscript. The idea that a high affinity protein will bind before a low affinity protein is self-evident, and the idea that if the high affinity protein is limiting that the low affinity protein may later have more total bound (e.g. if low-affinity is much more abundant than high affinity protein) is not novel. It isn't clear that this concept is needed or necessary to explain the author's data. Moreover, it is not clear if this concept is needed for the authors claim of perfect correlation between model predictions and PPIN observations in Fig 5B. If there is something significant here, the authors should explain it more thoroughly and more clearly. If there is not something here, the authors may wish to cut it from the manuscript.

We apologise for not explaining this better. The main point of the model is that the changes in the concentration of active RAS translates into a substantial change of interactors rather than just different strengths of signalling. While we agree that it is self-evident that high affinity interactors bind first, the differential binding of high versus low affinity RAS interactors is mainly a function of an increase in the abundance of active RAS. To the best of our knowledge this has not been shown before. In the context of this study, the important point this model makes is that the 3fold difference in active KRAS will substantially change the landscape of high vs. low affinity interactors (indicated by broken lines in Fig. 5A). This change in interactors plausibly could explain the source for the widespread network rewiring observed in the two cell lines as indicated by the strong correlation between model predictions and PPIN observations in Fig 5B. We have revised this paragraph to make the role of the model clearer and have summarised the conclusion from the model analysis as follows: "Thus, changes in mtKRAS concentration can profoundly rearrange the composition of KRAS-effector complexes, which rather than changing the strengths of downstream pathway activation shifts signaling from high to low affinity effectors as mtKRAS dosage increases (Fig. 5A)." Thus, we prefer to leave the model in, but hope we now have better explained the rationale. Please note that Fig. 5B now is Supplementary Table 13, because the journal's formatting guidelines do not allow tables in figures.

Smaller, specific, comments:

a) Can the Genomic data be used to determine precisely where the three claimed KRAS alleles in HKE3 are located? It would be valuable to better define the KRAS genomic situation.

We have tried extensively to map the localisation of the three KRAS alleles. The mapping is hampered by a high content of repetitive sequences in the KRAS locus and by the subtlety of the change. The disruption vector is a 7kb insertion consisting mainly of homologous KRAS sequences derived from a lung cancer cell line³. Thus, it is very difficult to map the precise genomic location. However, we have performed extensive PCR experiments on the genomic KRAS DNA and mRNA encoded by these loci in order to determine the underlying genetic cause for the difference in mtKRAS expression between HCT116 and HKE3 cells. The salient experiments are shown in Fig. S1A. They show the presence of three different types of KRAS mRNAs in HKE3, one encoded by the wildtype KRAS allele, one encoded by a mutant KRAS allele disrupted by the disruption vector, and one intact mutant KRAS allele without disruption.

b) The language of this paper was too strong. For example: page 4 "... inhibiting downstream effectors has proven ineffective due to complex feedback structures... and to the large number of effector pathways." The term "Proven" is quite strong. Perhaps, "... inhibiting downstream effectors has to date been ineffective, likely because of ..."

We had chosen this language for clarity but agree that the reviewer's suggestion is better. We have incorporated his/her phrase in the revised paper.

c) More problematic is the use of overly strong and definitive language to describe the data presented. Due to the problematic nature of HCT116 and HKE3, the authors need to be much more cautious with their interpretations and conclusions.

As stated above, we had chosen this language for clarity but have taken care to down tune the conclusions throughout the revised paper.

d) On pg 7 – the authors state their findings suggest many PPIs may be highly dependent on the cellular context. If true (and it likely is true), that would also suggest these results may not apply to other cancer cells, which have very, very, different sets of genomic and epigenetic alterations.

While it is correct and was surprising that 30% of all PPIs were changed between to isogenic cell lines, 70% of PPIs were conserved. This raises the interesting question how conserved the PPI changes are in mtKRAS cells. Therefore, we have added the following sentences to the discussion: "It also will be important to test whether the observed PPI rewiring is common to different mtKRAS cancer types. Our findings that PPI changes correlate with CRC patient prognosis and often affect proteins that are synthetic lethal with KRAS G13D³¹ indicate that a core signature of consistently altered PPIs may exist in mtKRAS cells."

e) Table S6. The authors state "only one expressed gene (NAV1) exhibited a non-synonymous variant predicted to alter protein function (Table S6)" The version of Table S6 that I have access to only lists one gene, NAV1. Were there non-synonymous variants NOT predicted to alter protein function? They should be listed as well.

As requested, we have now listed all synonymous and non-synonymous variants in the new Table S6.

It should be more clear in the table which method(s) were used to predict function, and what those algorithms predicted.

We have now included brief methods in the legends to the Tables as textboxes in the spreadsheets. The detailed methods are described in the Methods and Supplementary Methods.

Multiple approaches should be used to support the claim that only one needs to be considered.

As explained above, we have re-analysed the WGS data with the Broad Institute's GATK pipeline and Ensembl Variant Effect Predictor instead of snpEff, which we had used in the original manuscript. The new analysis is less stringent and identifies more SNVs, and we have included it in order detect any possible contributions of such genetic variations to the PPIN rewiring. Considering that the EGFR PPI networks contain 4,420 nodes, of which 1,360 have rewired interactions, SNVs affect 1.6% of nodes and 2.6% of rewired interactions. These data suggest that SNVs may be linked to the rewiring of a small number of EGFR network nodes, but cannot explain the extensive rewiring of 30.7% PPIs observed. This finding is consistent with our original conclusion that SNVs (other than the KRAS G13D mutation) have no major role in EGFR PPIN rewiring.

Whether other changes (copy number changes, splice-site changes, silent mutations that may shift codon usage to a more or less common tRNA, etc.) should be discussed, too. There is much more to genomics and cancer genomics than SNVs.

Our new analysis considers the impacts of structural variants, CNVs, missense variants, frameshift variants, splice region variants, splice donor and splice acceptor variants, stop codon gained variants, in frame insertions and deletions, lost start sites, and protein coding altering variants. However, we cannot rule out that these or other mutations indirectly affect PPIs by changing gene/protein expression. However, as shown in Fig. 3, we did not find a strong association between changes in gene/protein expression and PPI rewiring, suggesting a limited influence on PPI rewiring. We have

now discussed these possibilities in the text as follows: “Nonetheless, we cannot rule out that these or other genetic differences influence some PPIs by affecting gene promoter usage, mRNA editing, or codon usage. We also considered that rewired prey could simply represent lowly or highly expressed nodes. However, we found no bias in the gene expression distribution of rewired nodes compared to unchanged nodes (Supplementary Figure 5B) suggesting that genetic changes that alter gene/protein expression, e.g. CNVs, do not make major contributions to PPIN rewiring.”

f) TCGA analysis for survival. It is unclear if multiple hypothesis testing was properly controlled for. That is, did the authors originally say “we will use exactly the top 20 genes, look at only 5 yr survival, and determine a p-value.” Or did they consider multiple timepoints and multiple lists of genes of varying lengths.

We did not consider multiple timepoints or multiple lists of genes of varying lengths in the survival analysis – the decision to examine the top 20 most rewired baits was made *a priori*. Therefore, correction for multiple hypothesis testing is not required here. Furthermore, as reported in the paper we performed several different analyses to confirm that the association with patient survival is robust.

g) Mathematical model. The version of the supplementary materials available to this reviewer had the top and bottom of the equations cutoff. A reformatted version would be needed to evaluate their model.

We apologise for this formatting error. A correctly reformatted version is provided.

h) Figure 5B seems to be important, but it is not clear what is being presented. Are these predictions based on both abundances and Kd values?

The predictions are based on the outputs of the mathematical model, which calculates the fold change in KRAS-effector complex depending on active KRAS concentration. This is stated in the Supplementary Methods “The model allows us to rank KRAS binding partners according to the fold-changes in KRAS-bound fractions with the change in mtKRAS. The fold-changes equal RX_i^{hi}/RX_i^{lo} ratios (Eq. 1), where RX_i^{hi} and RX_i^{lo} are the concentrations of ith partner complex with KRAS for high and low doses of mtKRAS, respectively.” To clarify this we now describe these results in Supplementary Table 13 as follows: “Ranking of 8 bona-fide KRAS effector pathways according to the fold changes in the relative abundance of the respective KRAS-effector protein complexes as mtKRAS dosage changes.” Please note that the table originally presented in Fig. 5B is now Table S13, as according to formatting guidelines figures should not contain tables.

From what source did the Kd and abundance values come? The values used should be presented in a table.

These values and the sources they came from were presented as a table on p.11 of the Supplemental Methods. For convenience, the table is pasted below.

Binding partner	K_d value	Reference
RAF1	0.08 μ M	4
RALGDS	1.3 μ M	5
RASSF5	0.4 μ M	6
AFDN	3.0 μ M	7
RIN1	0.022 μ M	8
TIAM	1 μ M	No data, estimated
PLC ϵ	0.82 μ M	9
PI3K	204.7 μ M	4

There is no consensus in the literature what the values are (e.g. wide ranges of Kd values can be found in the literature for specific interactions). Are the predictions robust to the variation that is in the literature? How robust is the “perfect correlation”?

The overall ranking in the table shown in Fig. 5B (Supplementary Table 13) is robust over a range of input values that is mostly within the variations reported in the literature¹⁰. For low affinity binders the fold-changes RX_i^{hi}/RX_i^{lo} depend mainly on Kd values. Consequently, the position in the rank table for specific binding partner will change if its Kd became lower or higher than Kd of its neighbour in the table. For high affinity binders the fold-changes RX_i^{hi}/RX_i^{lo} and the position in the rank table are determined by both protein abundance and Kd. However, we have checked that at least 2-fold changes in protein abundance and Kd do not change the position of high-affinity binders in the rank table. Only 5fold or bigger changes of protein abundance or Kd may result in binding partners swapping places with its neighbours in the rank table. Thus, even with larger variations in input parameters the top 10 KRAS-effector complexes would still be the same.

How much of the correlation is only due to Kd, how much requires abundance? Etc.

As explained in the text on p.14 “This model classifies KRAS effectors into low and high affinity binders, whose binding dissociation constants (Kd’s) are greater or smaller, respectively, than the abundance of active KRAS. It shows that for low-affinity effectors the corresponding KRAS complex concentrations are proportional to the effector concentration divided by the Kd, whereas for high-affinity interactors the resulting KRAS complexes concentrations are determined by the abundance of active KRAS and effectors alone.”

i) The Discussion feels a bit generic re: the value of defining the PPINs. Does this study help move the field forward? How so? Are data & findings incremental or are they significant? If significant, please explain how so. Did this work uncover any important problems that should be fixed in future studies? A discussion of whether the differences between HKE3 and HCT116 were known before the study, or were discovered after the fact, could be valuable and help the field move forward and/or better anticipate similar problems.

We have revised the discussion according to these points.

Major suggestions:

Overall, this manuscripts reads as an impressively large and thorough body of work. It seems like almost every methodology to study the data was attempted. However, the here discovered and reported differences between HKE3 and HCT116 make it much less well-suited for PPIN comparison than the pair would have seemed before this new information became available.

It is not clear how this can be reasonably addressed. An “add back” of KRAS G13D into HKE3 cells could be utilized to test whether interactions specific to HCT116 or HKE3 are flipped with the reintroduction of G13D. However, as RAS activation is not either/or but is graded with thresholds (per the authors text), this experiment comes with numerous challenges. Alternatively, Shirasawa et al appear to have developed other isogenic versions of HCT116 that did not have KRAS G13D. If multiple G13D k/o cells have similar PPINs, that would support that what is being observed is truly due to the decrease in KRAS G13D expression and not due to other, random, unshared, events. Lastly, cancer synthetic lethal experiments, often utilize sets of KRAS mutant and KRAS WT cells to find the changes that are “universal” for Mutant and WT cells and to compare differences. Validation in additional KRAS mutant/WT isogenic (or mostly) isogenic pairs could be valuable. Of course, these 3 sets of experiments are likely cost and/or time prohibitive. They are discussed simply to put the flaws of this study in context and to contribute to discussion of how to design better studies going forward.

We agree with the reviewer that such experiments would be valuable but are cost and time prohibitive. As suggested by the reviewer we have discussed them as follows: "The influence of such factors could be addressed by reconstitution experiments that titrate mtKRAS dosage and by studying other isogenic cell line pairs." Unfortunately, space constraints do not allow a more thorough discussion.

The most important changes, in my opinion, are clear explanations of the limitations of the study and the use of much less strong language to describe the findings.

The other issue with this manuscript is it is not clear what new understanding of biology comes from the manuscript. That there are differences and "rewiring", etc., seems logical and consistent with many other studies, even if this exact study was not previously done.

We agree that a clear explanation of the limitations of the study is important and have included this in the discussion. However, we are not aware of many other studies that demonstrate rewiring of PPIs except a few studies that looked at PPI rewiring in isolated cases of protein complexes. In particular, we are not aware of studies showing that a mutation in a protein can lead to widespread PPI rewiring throughout a PPI network far beyond local (i.e. direct) interactions. Regarding what new biological understanding comes from our work, the reviewer summarises this nicely in his/her opening paragraph "... the work investigates how oncogenic KRAS promotes cancer by means of how signals propagate downstream from it, which is a long-standing, well-studied, and still incompletely understood problem."

Reviewer #2 (Remarks to the Author): Expert in proteomics

The manuscript by Kennedy and co-workers describes a systematic analysis of protein-protein interactions (PPIs) in two colorectal cancer cell lines that were designed to differ in the levels of oncogenic KRAS expression. To this end, the authors perform quantitative affinity purification mass spectrometry experiments. They observe extensive rewiring of the PPI network, including nodes that are far away from the immediate impact of KRAS. These perturbations appear to be relevant for CRC since they correlate with survival.

This paper addresses an exciting and highly relevant biomedical question and presents an impressive amount of data. The proteomic data appears to be of high quality and the described data analysis pipeline is sound. The modeling part also sounds convincing, although I am not able to evaluate the details of this. Overall, I think this paper will become suitable for publication when the authors have addressed a number of important points:

1. Page 6: "The baits were expressed as FLAG-tagged proteins carefully titrating transfection to achieve a modest and similar level of overexpression in both cell lines." The endogenous expression levels of the bait proteins is different. Therefore, titrating transfections to achieve "modest and similar overexpression" requires experiments such as western blots with antibodies against the bait protein (not the FLAG tag) to compare the level of exogenous and endogenous protein. Otherwise, the degree of overexpression will be modest for more abundant proteins but very high for less abundant proteins. I cannot find such western blot data. Either such data should be provided or the statement has to be reworded to indicate that the degree of overexpression is not always modest.

We have assessed overexpression for many but not all baits due to lack of high-quality antibodies suitable for Western blotting for some baits. An example is shown in Fig. A below. However, we

concede the reviewer's point and have reworded the statement that "...FLAG-tagged proteins carefully titrating transfection to achieve a similar expression level in both cell lines."

Fig. A. Quantification of the expression of a Flag-tagged bait (A-Raf) in HCT116 and HKE3 cells. Cell lysates were Western blotted with A-Raf and Actin (loading control) antibodies, and quantified with Image J. After normalisation to the loading control, the signals for the endogenous A-Raf proteins were subtracted from the signal of the lanes containing overexpressed Flag-A-Raf. The results showed a 1.59 and 1.75 fold overexpression of Flag-A-Raf, respectively.

2. Related to the point above, it should be mentioned that network rewiring could also occur as a result of differential bait expression. In fact, this is exactly what the authors observe for KRAS. Since this study used exogenous baits that were artificially overexpressed at a similar level in both cell lines, this aspect of network rewiring cannot be addressed.

The reviewer is correct. However, according to the protein expression profiling data only 2 baits were differentially expressed endogenously: RPS6KA1 (FC = -1.7) and SH3KBP1 (FC = -1.7) showed modestly lower protein abundance in HCT116 cells. Furthermore, more globally we assessed the association between differential protein expression and PPI rewiring and found only a very weak correlation ($r^2 = 0.18$). These data indicate that differences in protein (or bait protein) expression have a limited association with network rewiring. KRAS was not a bait in our analysis.

3. While the authors validate their AP-MS data, I don't see validation for the membrane yeast two-hybrid data (MYTH) that they also report. The overall question is how sensitive and specific the MYTH data is. More generally, it is not clear to me if/what the MYTH data really contributes to this paper, especially since it cannot reveal rewiring of PPIs in response to oncogenic KRAS. Is it really worth to include this data in the paper at all?

As AP-MS is known to underrepresent interactors of integral membrane proteins, MYTH was used to identify binary protein interactors of the human ERBB family. While the reviewer is correct that this analysis cannot reveal rewiring of these interactions, this analysis does identify 181 new ERBB interactions, a valuable resource for the community. We also now show data that the interactions found by MYTH are linked to different diseases and tissues (included in an extended Table S3). While we did not perform a validation by co-IP/WB for the MYTH data in this study, our previous results using MYTH to map the interactome of ERBB1¹¹ showed that ca. 83% of MYTH interactions (15 out of 18 tested) could be verified by co-IP/WB experiments in mammalian cells.

4. Page 10: The authors find a significant correlation between between differential node abundance and network rewiring. From this, they conclude that "differences in protein expression between mtKRASHi and mtKRASLo cells can explain some of the rewired interactions.". I agree with this conclusion. In addition, the authors should also consider the alternative possibility that rewiring the interactions causes differences in protein abundance: A number of studies have shown that complex formation can stabilise proteins and thus increase steady-state protein levels

(<https://doi.org/10.1016/j.cell.2016.09.015>; <https://doi.org/10.15252/msb.20177548>;

<https://doi.org/10.1074/mcp.M116.060301>). This alternative possibility should at least be mentioned.

Good point. We have mentioned this possibility in the text: “Furthermore, complex formation can stabilize proteins¹² and may contribute to the differential protein abundance between mtKRAS^{Hi} and mtKRAS^{Lo} cells.”

The authors may also want to include a more detailed analysis using their mRNA data for both cell lines: If differential node abundance is only seen at the protein but not at the mRNA level, this suggests that differential interaction causes differences in node abundance (and not vice versa). A more detailed analysis of the RNAseq data has been published recently by us in the British Journal of Cancer (<https://www.nature.com/articles/s41416-019-0477-7>). We felt that the current manuscript already has too much data in it for a sufficiently detailed analysis of the RNAseq data to be presented. However, as per the reviewer’s suggestion we have assessed whether genes that were found to be differentially expressed between the HKE3 and HCT116 cell lines were significantly enriched among rewired nodes. While more than 40% of the rewired nodes showed evidence of differential gene expression (FDR < 0.05), rewired nodes were not statistically enriched for differentially expressed genes compared to non-rewired nodes (see calculations below).

Association between DE genes and PPI rewiring :	
Enrichment for differentially expressed genes among rewired preys	
Total proteins in EGFRNet:	1638
Differentially expressed:	660
Total number of rewired proteins in the EGFRNet:	735
Number of rewired proteins that are differentially expressed:	310
P(X>=310)	0.08826 (not significant)

It must be noted however that we do not feel these data are sufficient to support the conclusion that this means that rewiring drives differences in node abundance (and not vice versa), though we certainly recognise this is a possibility and have now mentioned this in the revised manuscript.

5. Figure 5C: According to the text this is a volcano plot. I don’t fully understand this: A typical volcano plot displays the -log significance on the y axis. This plot shows the IF score ratio instead. Not unexpectedly, this ratio tends to be more extreme in the region where the IF score is close to 0 (i.e. data points around the y axis). Thus, some IF score ratios are huge although the absolute IF score is very small, which is why the proteins in this region are probably not very informative. In a typical volcano plot this region would be depopulated (proteins which do not change are also not significant).

We apologise for this confusion. We have borrowed the idea of a volcano plot for Fig. 5C, but it is not a volcano plot in the traditional sense, where fold change is plotted vs. significance. We plot the IF score for a node in a cell line vs the ratio of IF scores between corresponding nodes in the two cell lines. That means nodes close to the y-axis receive less information flow, but this small flow still can mark an important difference in signal flux between the cell lines if the IF ratio is high. To avoid confusion we have dropped the name volcano plot, and have coloured only the nodes that were in the top 5th percentile in terms of their information flow score and were predicted to receive more flow in either mtKRAS^{Lo} or mtKRAS^{Hi} cells. Please note that former Fig. 5C is now Fig. 5B in the revised paper.

6. Impact of phosphorylation on PPIs: Several interactions downstream of EGFR are well-known to be regulated by tyrosine phosphorylation via SH2 domains. I am missing a more detailed analysis of

this: Do the authors see a correlation between differential tyrosine phosphorylation and corresponding PPIs?

This is a good suggestion. Unfortunately, we found that only 2 tyrosine phosphorylation sites are significantly different between the two cell lines. Only one of these tyrosine phosphorylation sites (in UBR4) is weakly predicted to bind to SH2 domains (only picked up by Scansite 4.0 at minimum stringency). However, UBR4 has no differential protein interactions of with proteins containing SH2 sites (as predicted by Scansite 4.0 at minimum stringency).

7. The authors mention a number of different mechanisms that lead to network rewiring. It would be great to have an estimate how important each of these mechanisms is. Can this somehow be estimated based on the data?

Figure 3A and 3B show the proportions of rewired preys for each bait that are differentially abundant or differentially phosphorylated, respectively. As can be seen in this analysis the association is variable for different baits, but on average can account for no more than 17% of the rewiring. It should be noted that we cannot infer a causal relationship between differential abundance/phosphorylation and rewiring – just a statistical association. The statistical analysis is already discussed in detail in the section of the manuscript entitled “Drivers of PPIN rewiring”.

8. Supplemental material Table A: The last line of the previous paragraph is inserted between the table and the legend. This should be fixed.

This has now been corrected.

9. Supplemental material: The authors discuss the benefits of SILAC label swapping/switching. I agree with them that label swapping is useful, but I disagree with the statement that “label switching is not commonly done in SILAC experiments”. It is actually done routinely in many labs, so this idea is not new. The authors should also stick to one term. I would prefer label swapping over label switching - this is also the term that is typically used in the proteomics community.

We could not agree more with the reviewer. This statement stems from a previous review of this paper at Science where one reviewer kept insisting that label switching is uncommon and unorthodox in proteomics experiments and compromises the experimental design. We have removed this statement and have changed “switching” back to “swapping”.

10. Supplemental Material: The authors write that they used a “high-resolution pre-scan” with two microscans per spectrum. I don’t understand what this means. Orbitrap instruments typically use prescans for automatic gain control (AGC), but I am not aware of an option to use two such prescans. Also, it appears that no MS1 scans were made. I assume that the authors did not use prescans but full (MS1) scans at R=30,000? Is this indeed the case? If yes, why did they decide to record two microscans? A single full scan at a resolution of R=60,000 would have given better data (higher resolution) with the same duty cycle.

We did not perform a high resolution pre-scan, this is actually the MS1 scan. We thank the reviewer for picking up this error and have corrected it in the Supplemental Material as follows: “From the MS1 scan (resolution: 30 000, two microscans per spectrum) with a mass range of 300–1,500 the 10 most intense peptide ions were selected for fragment analysis ...”

The description of the two microscans per spectrum is correct. In the “Define Scan” menu of the Orbitrap Velos Tune, one can set the number of microscans that is used (Fig. B). Both scans will then be combined and result in a single spectrum. Using the two microscan method gave us slightly, but reproducibly better results, possibly because the precursor selection and AGC uses the first microscan for selecting ions and calculating fill times.

Fig. B. Screenshot of the “Define Scan” menu of the Orbitrap Velos

11. While I am not able to check all details of the “HiQuant” method they used, the data analysis strategy is sound and well described. I only have one question here: Why did the authors use “significance A” rather than “significance B” as their second filter criterion? Significance B has the additional advantage that protein abundance is taken into account. This may be useful, since SILAC ratios of lower abundant proteins tend to show a higher variability. In other words, significance A is expected to favour low abundant proteins and to bias against high abundant proteins in the pull-downs.

In our experience, to apply significance B efficiently, Maxquant searches need to be run separately for each bait, and intensities extracted for each experimental condition. Such an application is easily possible for a small number of baits / conditions, but would be very tedious in our study, where we compared 2,280 conditions (95 baits with their controls in 2 cell lines, forward and reverse SILAC labeling in 3 biological replicates each). Thus, we used significance A and applied extra stringent filtering detailed in the method section and in the HiQuant paper (a p-value of 0.05 on a t-test filtering on the SILAC ratios between biological replicates, followed by the Significance A calculation). Fig S5A shows that there is no difference in gene expression of rewired preys versus all/unchanged preys confirming that rewired preys are not biased towards lowly abundant expression.

This review is from Matthias Selbach

Reviewer #3 (Remarks to the Author): Expert in colorectal cancer

Susan Kennedy, Walter Kolch and colleagues have used quantitative mass spectrometry (qMS) to map KRAS specific changes in protein-protein interactions in two isogenic CRC cell lines that express either high or low levels of mutated KRAS. They found interesting rewiring of the signalling network as an effect following mutations in KRAS. Even though the study has many strengths, it has also some major shortcomings.

The authors have based their study on colorectal cancer (CRC) cells expressing transforming (high) levels of KRASG13D (mtKRAS). Unfortunately, the cells HCT116 and MKE are both of MSI (microsatellite instable) phenotype. Those cancer cells are known to have specific characteristics compared to non-MSI tumors. MSI tumors stands for only about 15% of CRCs and can therefore

reflect colorectal cancer on general terms. The specific molecular nature of MSI tumors might have had impact on the given results.

We agree that the MSI phenotype could potentially influence some of our findings. Ideally, we would have performed this analysis with a panel of cell lines, but then the scope and depth of analysis presented in our paper would not have been possible. Faced with this judgment call we chose to carry out a deep and comprehensive analysis in a cell line pair rather than a more limited analysis across many cell lines. We have used HCT116 cells because (i) they are a widely used model for CRC cells explicitly mentioned in 11,645 Pubmed abstracts; (ii) they have isogenic derivative cell lines available where mtKRAS was reported to be knocked out; (iii) these isogenic pairs have been widely used in synthetic lethal screens; and (iv) that HCT116 are now also being used by the Bioplex team, which aims to globally map the human protein interactome (https://www.youtube.com/watch?v=KzqY6UvyN_g). Thus, of all the imperfect choices available, HCT116 seemed to be a sensible one.

Even more problematic, HCT116 and MKE harbour a not only a KRAS G13D mutation but also a PIK3CA mutation, protein involved in the same signalling network. This can of course affect the cellular response and specific protein-protein interactions. One way of put light on this problem would be to include more cell lines with different molecular characteristics, but this is of course probably a very different paper in the end.

We have performed whole genome sequencing of both cell lines. The results, briefly mentioned in the main text and shown in Table S6, clearly demonstrate that HCT116 and HKE3 have the same mutational landscape (including the same PIK3CA mutation). Thus, while it is plausible that these other mutations impact PPIN, they are the same in the two cell lines and hence well controlled for.

The survival analyses using the TCGA data set were analysed without taken other molecular and clinical data into account. Why were not MSI cases analysed separately? This would be specifically important taken their cell lines with MSI status into account. Is the mild prognostic effect seen in both MSI and non-MSI cases? How is stage distributed among the two groups in the survival analyses? I would suggest to also include a multivariable Cox regression survival model in order to understand the prognostic information better. It can be somewhat misleading to only include univariable analyses.

Unfortunately, TCGA does not provide the MSI status for the multi-omic CRC dataset used in our analysis. Therefore, this analysis was not possible. In order to assess whether tumour stage influences the results, we used the cBioportal group comparison feature (<https://www.cbioportal.org/tutorials#group-comparison>) to statistically compare the clinical metadata of TCGA CRC patients with and without alterations in the top 20 most rewired baits. The cohort showed no statistically significant difference in American Joint Committee on Cancer Tumor Stage Code (P=0.44); American Joint Committee on Cancer Metastasis Stage Code (P=0.06); or Neoplasm Disease Stage American Joint Committee on Cancer Code (P=0.2). Including these staging parameters into a multivariable Cox regression analysis showed that the model based on EGFR network node rewiring is still statistically significant when adjusted for metastasis stage or disease stage. These results indicate that node rewiring is an independent prognostic parameter for disease stage and distant metastasis. These data have now been included into Figure S9:

D

Multivariate COX regression analysis					
Top 20 rewired baits with genetic alterations	coef	exp(coef)	se(coef)	z Pr(> z)	p-value
adjusted for metastasis stage:	0.8117	2.2516	0.4094	1.983	0.0474
Top 20 rewired baits with genetic alterations	coef	exp(coef)	se(coef)	z Pr(> z)	p-value
adjusted for disease stage:	9.619e-01	2.62E+00	4.378e-01	2.197	0.028

Fig. S9 (D) Multivariate COX regression for the TCGA patients was calculated in R.

Moreover, G13D is stated in the introduction to be the second most common specific KRAS mutation in CRC, and reference no. 7 is given. I would suggest the authors to carefully check this statement.

We have checked this statement and the summary figure in the publication (Ref. 7) confirms this statement. This statement is also confirmed by data from The Cancer Genome Atlas (TCGA):

http://www.cbioportal.org/results/mutations?Action=Submit&cancer_study_list=coad_caseccc_2015%2Ccoadread_dfc_2016%2Ccoadread_genentech%2Ccoadread_tcga_pub%2Ccoadread_tcga_pan_can_atlas_2018%2Ccoadread_tcga%2Ccoadread_mskcc%2Ccrc_msk_2017&case_set_id=all&data_priority=0&gene_list=KRAS&tab_index=tab_visualize

References

- 1 McLaren, W. *et al.* The Ensembl Variant Effect Predictor. *Genome Biol* **17**, 122 (2016).
- 2 Matallanas, D. *et al.* Mutant K-Ras activation of the proapoptotic MST2 pathway is antagonized by wild-type K-Ras. *Mol Cell* **44**, 893-906 (2011).
- 3 Shirasawa, S., Furuse, M., Yokoyama, N. & Sasazuki, T. Altered growth of human colon cancer cell lines disrupted at activated Ki-ras. *Science (New York, N.Y.)* **260**, 85-88 (1993).
- 4 Nakhaeizadeh, H., Amin, E., Nakhaei-Rad, S., Dvorsky, R. & Ahmadian, M. R. The RAS-Effector Interface: Isoform-Specific Differences in the Effector Binding Regions. *PloS one* **11**, e0167145 (2016).
- 5 Linnemann, T., Zheng, Y.-H., Mandic, R. & Matija Peterlin, B. Interaction between Nef and Phosphatidylinositol-3-Kinase Leads to Activation of p21-Activated Kinase and Increased Production of HIV. *Virology* **294**, 246-255 (2002).
- 6 Harjes, E. *et al.* GTP-Ras Disrupts the Intramolecular Complex of C1 and RA Domains of Nore1. *Structure* **14**, 881-888 (2006).
- 7 Rudolph, M. G. *et al.* Thermodynamics of Ras/Effector and Cdc42/Effector Interactions Probed by Isothermal Titration Calorimetry. *Journal of Biological Chemistry* **276**, 23914-23921 (2001).
- 8 Wang, J., Peng, X., Li, M. & Pan, Y. Construction and application of dynamic protein interaction network based on time course gene expression data. *Proteomics* **13**, 301-312 (2013).
- 9 Wohlgemuth, S. *et al.* Recognizing and Defining True Ras Binding Domains I: Biochemical Analysis. *Journal of Molecular Biology* **348**, 741-758 (2005).
- 10 Fujioka, A. *et al.* Dynamics of the Ras/ERK MAPK cascade as monitored by fluorescent probes. *The Journal of biological chemistry* **281**, 8917-8926 (2006).
- 11 Deribe, Y. L. *et al.* Regulation of epidermal growth factor receptor trafficking by lysine deacetylase HDAC6. *Sci Signal* **2**, ra84 (2009).
- 12 McShane, E. *et al.* Kinetic Analysis of Protein Stability Reveals Age-Dependent Degradation. *Cell* **167**, 803-815.e821 (2016).

REVIEWERS' COMMENTS:

Reviewer #1 (Remarks to the Author):

The authors have done a good bit of work to address the original critiques. I think most of the remaining differences that I have with the conclusions of the paper are opinion, and that the authors have done an appropriate job of communicating their studies and findings.

My only new concern has to do with the recent Charitou et al British Journal of Cancer study from the same authors. It has similar conclusions, and it was not cited. I think there are some significant issues that must carefully be addressed to avoid appearances of double-publication of the same data, etc.

For example, both papers cite RNAseq expression data under GSE105094 in the Gene Expression Omnibus. It therefore appears to be the same data being analyzed in both studies. I would think it is more appropriate for one study to originally present the data (now, the Charitou et al ms) and then the following study (the present study) to cite the earlier work and state that they are using the same data.

Additionally, that the earlier paper was not cited in the manuscript seems troublesome as the conclusions of extensive rewiring are shared. They were distinct methods (RNA seq vs proteomics.) However, it seems logical to assume that rewiring in one would imply rewiring in other (e.g. if PPI are rewired, one would expect transcriptional consequences, and if gene expression is significantly different with different pathways up & down regulated, one would expect PPI rewiring). This begs the question of whether the authors "scooped" themselves with their other publication.

At minimum, the authors should

- a) Cite their Charitou et al manuscript
- b) Clearly indicate which data sets they are analyzing here were previously published in the Charitou (or any other) publications
- c) Add a discussion that discusses how the current study complements and/or builds on the other study

Reviewer #2 (Remarks to the Author):

The authors have appropriately addressed all comments. I think the paper can now be accepted.

Reviewer #3 (Remarks to the Author):

I have read the resubmitted material including the rebuttal letter. I find that the manuscript has improved its overall scientific quality and has many advantages.

However, I still have one major point that I think needs to be handled with. The survival analysis of the TCGA CRC patients with and without alterations in the top 20 most rewired baits has also been improved in the resubmitted version. Nevertheless, the multivariable Cox regression model is modelling the overall risk of Death and not cancer-specific deaths. In that perspective the authors should take patient age into account. I would suggest to expand the Cox model with not only AJCC tumor stage but also patient age, and also localization in the colon and rectum. A survival analysis of overall survival without age is not an appropriate analysis. Colon and rectal cancers have somewhat different clinical characteristics and colon and rectum cases can preferentially be separated. Furthermore, from my point of view the Cox models should only include the full AJCC

stage (as categorical variable) and not the variable "AJCC Metastasis Stage", which might be interesting but only in a more detailed analysis.

A minor point: The MSI status is still very interesting in understanding the main findings in this manuscript. I am aware of the problem having the MSI data out from the TCGA network, but it is included as one out of many variables in the original TCGA Nature-publication from 2012 on colorectal cancer.

Response to Reviewers' Comments

(Our responses are in blue)

Reviewer #1 (Remarks to the Author):

The authors have done a good bit of work to address the original critiques. I think most of the remaining differences that I have with the conclusions of the paper are opinion, and that the authors have done an appropriate job of communicating their studies and findings.

My only new concern has to do with the recent Charitou et al British Journal of Cancer study from the same authors. It has similar conclusions, and it was not cited. I think there are some significant issues that must carefully be addressed to avoid appearances of double-publication of the same data, etc.

For example, both papers cite RNAseq expression data under GSE105094 in the Gene Expression Omnibus. It therefore appears to be the same data being analyzed in both studies. I would think it is more appropriate for one study to originally present the data (now, the Charitou et al ms) and then the following study (the present study) to cite the earlier work and state that they are using the same data.

Additionally, that the earlier paper was not cited in the manuscript seems troublesome as the conclusions of extensive rewiring are shared. They were distinct methods (RNA seq vs proteomics.) However, it seems logical to assume that rewiring in one would imply rewiring in other (e.g. if PPI are rewired, one would expect transcriptional consequences, and if gene expression is significantly different with different pathways up & down regulated, one would expect PPI rewiring). This begs the question of whether the authors "scooped" themselves with their other publication.

At minimum, the authors should

- a) Cite their Charitou et al manuscript
- b) Clearly indicate which data sets they are analyzing here were previously published in the Charitou (or any other) publications
- c) Add a discussion that discusses how the current study complements and/or builds on the other study

We can assure the reviewer that there is no overlap between the current paper and Charitou et al. (Br J Cancer. 121:37-50, 2019). The Charitou paper used metabolomics and an extensive set of RNAseq data to analyse metabolomic and transcriptomic changes related to the expression of mutant KRAS. These data sets were generated in two cell line pairs: the HCT116 & HKE3 (i.e. the same cells used in the current paper), and HKe3 cells reconstituted with either wildtype KRAS or mutant KRAS^{G13D} (Tsunoda et al., Anticancer Res 35: 4453-59, 2015). The latter cells were not used in the current paper. In fact, it was experiments in this latter cell line pair that led to the salient conclusion of the Charitou paper: "While TGF α treatment strongly activated protein synthesis in wtKRAS cells, protein synthesis was not activated above basal levels in the TGF α -treated mtKRAS cells. This was likely due to the defective activation of the mTORC1 and other pathways by TGF α in mtKRAS cells, which was associated with impaired activation of PKB signalling and a transient induction of AMPK signalling." (cited from the abstract of Charitou et al.). The RNAseq data reported in Charitou et al. were obtained from time course treatments of all four cell lines at 0, 15, 30, 60, 90 and 120 min after stimulation with TGF α , and the corresponding data sets were deposited in the GEO database under accession numbers GSE105094 (HCT116 & HKE3) and GSE110649 (HKe3-wtKRAS & HKe3-mtKRAS). The Charitou paper reports the full analysis of both RNAseq datasets. In the current paper we have used the RNAseq data of 4 genes (FOXO1, MYC, STAT1, FOS) from the GSE105094 data set to validate that the signal flow to these transcription factor genes is different in

HKE3 and HCT116 cells. As only a tiny fraction of the GSE105094 data (4 of 58,051 mRNA species) was used and for a very different purpose, we can re-assure the reviewer that this is not a case of trying to double publish. The reason why the Charitou paper was not cited is simply due to the fact that it had not been published (or e-published) when we first submitted our manuscript to Nature Communications. To clarify the issue, we - as advised by the reviewer - have cited the Charitou paper, briefly discussed its findings (in the discussion of the paper) and clarified which part of the GSE105094 data set has been used (in the supplementary data). This data set has not been used by us in any other publications. However, as it is publically available, we cannot exclude that other parties have used it in publications.

Reviewer #2 (Remarks to the Author):

The authors have appropriately addressed all comments. I think the paper can now be accepted. We are pleased that the reviewer found our revisions satisfactory.

Reviewer #3 (Remarks to the Author):

I have read the resubmitted material including the rebuttal letter. I find that the manuscript has improved its overall scientific quality and has many advantages. However, I still have one major point that I think needs to be handled with. The survival analysis of the TCGA CRC patients with and without alterations in the top 20 most rewired baits has also been improved in the resubmitted version. Nevertheless, the multivariable Cox regression model is modelling the overall risk of Death and not cancer specific deaths. In that perspective the authors should take patient age into account. I would suggest to expand the Cox model with not only AJCC tumor stage but also patient age, and also localization in the colon and rectum. A survival analysis of overall survival without age is not an appropriate analysis. Colon and rectal cancers have somewhat different clinical characteristics and colon and rectum cases can preferentially be separated. Furthermore, from my point of view the Cox models should only include the full AJCC stage (as categorical variable) and not the variable "AJCC Metastasis Stage", which might be interesting but only in a more detailed analysis.

As requested by the reviewer we have repeated the survival analysis including patient age and AJCC disease stage in the Cox regression model. Including these parameters in the model led to a more significant P value ($P = 0.03$) compared to our original analysis ($P = 0.04$). These data show the survival analysis presented in the paper is robust to the factors identified by the reviewer as being potentially confounding. We also found that there was no significant difference in the age of patients with/without alterations in the top 20 bait proteins. These results are now mentioned in the manuscript.

A minor point: The MSI status is still very interesting in understanding the main findings in this manuscript. I am aware of the problem having the MSI data out from the TCGA network, but it is included as one out of many variables in the original TCGA Nature-publication from 2012 on colorectal cancer.

We agree that analysing MSI status would be interesting. The Supplementary Table 1 from the paper the reviewer refers to contains the MSI status for only 276 patients. Our analysis used the much larger (640 samples) colorectal cancer dataset available in TCGA. Unfortunately, this larger dataset does not have MSI status associated with it, and therefore we regrettably cannot perform the analysis suggested by the reviewer.